# How degenerate is the parametrization of neural networks with the ReLU activation function?

**Julius Berner**
Faculty of Mathematics, University of Vienna
Oskar-Morgenstern-Platz 1, 1090 Vienna, Austria
`julius.berner@univie.ac.at`

**Dennis Elbrächter**
Faculty of Mathematics, University of Vienna
Oskar-Morgenstern-Platz 1, 1090 Vienna, Austria
`dennis.elbraechter@univie.ac.at`

**Philipp Grohs**
Faculty of Mathematics and Research Platform DataScience@UniVienna, University of Vienna
Oskar-Morgenstern-Platz 1, 1090 Vienna, Austria
`philipp.grohs@univie.ac.at`

## Abstract

Neural network training is usually accomplished by solving a non-convex optimization problem using stochastic gradient descent. Although one optimizes over the networks parameters, the main loss function generally only depends on the realization of the neural network, i.e. the function it computes. Studying the optimization problem over the space of realizations opens up new ways to understand neural network training. In particular, usual loss functions like mean squared error and categorical cross entropy are convex on spaces of neural network realizations, which themselves are non-convex. Approximation capabilities of neural networks can be used to deal with the latter non-convexity, which allows us to establish that for sufficiently large networks local minima of a regularized optimization problem on the realization space are almost optimal. Note, however, that each realization has many different, possibly degenerate, parametrizations. In particular, a local minimum in the parametrization space needs not correspond to a local minimum in the realization space. To establish such a connection, inverse stability of the realization map is required, meaning that proximity of realizations must imply proximity of corresponding parametrizations. We present pathologies which prevent inverse stability in general, and, for shallow networks, proceed to establish a restricted space of parametrizations on which we have inverse stability w.r.t. to a Sobolev norm. Furthermore, we show that by optimizing over such restricted sets, it is still possible to learn any function which can be learned by optimization over unrestricted sets.

## 1 Introduction and Motivation

In recent years much effort has been invested into explaining and understanding the overwhelming success of deep learning based methods. On the theoretical side, impressive approximation capabilities of neural networks have been established [9, 10, 16, 20, 32, 33, 37, 39]. No less important are recent results on the generalization of neural networks, which deal with the question of how

well networks, trained on limited samples, perform on unseen data [2, 3, 5–7, 17, 29]. Last but not least, the optimization error, which quantifies how well a neural network can be trained by applying stochastic gradient descent to an optimization problem, has been analyzed in different scenarios [1, 11, 13, 22, 24, 25, 27, 38]. While there are many interesting approaches to the latter question, they tend to require very strong assumptions (e.g. (almost) linearity, convexity, or extreme over-parametrization). Thus a satisfying explanation for the success of stochastic gradient descent for a non-smooth, non-convex problem remains elusive.

In the present paper we intend to pave the way for a functional perspective on the optimization problem. This allows for new mathematical approaches towards understanding the training of neural networks, some of which are demonstrated in Section 1.2. To this end we examine degenerate parametrizations with undesirable properties in Section 2. These can be roughly classified as

C.1 unbalanced magnitudes of the parameters

C.2 weight vectors with the same direction

C.3 weight vectors with directly opposite directions.

Under conditions designed to avoid these degeneracies, Theorem 3.1 establishes inverse stability for shallow networks with ReLU activation function. This is accomplished by a refined analysis of the behavior of ReLU networks near a discontinuity of their derivative. Proposition 1.2 shows how inverse stability connects the loss surface of the parametrized minimization problem to the loss surface of the realization space problem. In Theorem 1.3 we showcase a novel result on almost optimality of local minima of the parametrized problem obtained by analyzing the realization space problem. Note that this approach of analyzing the loss surface is conceptually different from previous approaches as in [11, 18, 23, 30, 31, 36].

## 1.1 Inverse Stability of Neural Networks

We will focus on neural networks with the ReLU activation function $\rho(x) := x_+$, and adapt the mathematically convenient notation from [33], which distinguishes between the *parametrization* of a neural network and its *realization*. Let us define the set $\mathcal{A}_L$ of all network *architectures* with depth $L \in \mathbb{N}$, input dimension $d \in \mathbb{N}$, and output dimension $D \in \mathbb{N}$ by

$$\mathcal{A}_L := \{(N_0, \ldots, N_L) \in \mathbb{N}^{L+1} \colon N_0 = d, N_L = D\}. \tag{1}$$

The architecture $N \in \mathcal{A}_L$ simply specifies the number of neurons $N_l$ in each of the $L$ layers. We can then define the space $\mathcal{P}_N$ of *parametrizations* with architecture $N \in \mathcal{A}_L$ as

$$\mathcal{P}_N := \prod_{\ell=1}^{L} \left(\mathbb{R}^{N_\ell \times N_{\ell-1}} \times \mathbb{R}^{N_\ell}\right), \tag{2}$$

the set $\mathcal{P} = \bigcup_{N \in \mathcal{A}_L} \mathcal{P}_N$ of all parametrizations with architecture in $\mathcal{A}_L$, and the *realization* map

$$\mathcal{R} \colon \mathcal{P} \to C(\mathbb{R}^d, \mathbb{R}^D)$$
$$\Theta = ((A_\ell, b_\ell))_{\ell=1}^{L} \mapsto \mathcal{R}(\Theta) := W_L \circ \rho \circ W_{L-1} \ldots \rho \circ W_1, \tag{3}$$

where $W_\ell(x) := A_\ell x + b_\ell$ and $\rho$ is applied component-wise. We refer to $A_\ell$ and $b_\ell$ as the weights and biases in the $\ell$-th layer.

Note that a parametrization $\Theta \in \Omega \subseteq \mathcal{P}$ uniquely induces a realization $\mathcal{R}(\Theta)$ in the realization space $\mathcal{R}(\Omega)$, while in general there can be multiple non-trivially different parametrizations with the same realization. To put it in mathematical terms, the realization map is not injective. Consider the basic counterexample

$$\Theta = ((A_1, b_1), \ldots, (A_{L-1}, b_{L-1}), (0, 0)) \quad \text{and} \quad \Gamma = ((B_1, c_1), \ldots, (B_{L-1}, c_{L-1}), (0, 0)) \tag{4}$$

from [34] where regardless of $A_\ell, B_\ell, b_\ell$ and $c_\ell$ both realizations coincide with $\mathcal{R}(\Theta) = \mathcal{R}(\Gamma) = 0$. However, it it is well-known that the realization map is locally Lipschitz continuous, meaning that close[1] parametrizations in $\mathcal{P}_N$ induce realizations which are close in the uniform norm on compact

sets, see e.g. [2, Lemma 14.6], [7, Theorem 4.2], and [34, Proposition 5.1].

We will shed light upon the inverse question. Given realizations $\mathcal{R}(\Gamma)$ and $\mathcal{R}(\Theta)$ that are close, do the parametrizations $\Gamma$ and $\Theta$ have to be close? In an abstract setting we measure the proximity of realizations in the norm $\|\cdot\|$ of a Banach space $\mathcal{B}$ with $\mathcal{R}(\mathcal{P}) \subseteq \mathcal{B}$, while concrete Banach spaces of interest will be specified later. In view of the above counterexample we will, at the very least, need to allow for the reparametrization of one of the networks, i.e. we arrive at the following question.

> Given $\mathcal{R}(\Gamma)$ and $\mathcal{R}(\Theta)$ that are close, does there exist a parametrization $\Phi$ with $\mathcal{R}(\Phi) = \mathcal{R}(\Theta)$ such that $\Gamma$ and $\Phi$ are close?

As we will see in Section 2, this question is fundamentally connected to understanding the redundancies and degeneracies of the way that neural networks are parametrized. By suitable regularization, i.e. considering a subspace $\Omega \subseteq \mathcal{P}_N$ of parametrizations, we can avoid these pathologies and establish a positive answer to the question above. For such a property the term *inverse stability* was introduced in [34], which constitutes the only other research conducted in this area, as far as we are aware.

**Definition 1.1** (Inverse stability)**.** *Let $s, \alpha > 0$, $N \in \mathcal{A}_L$, and $\Omega \subseteq \mathcal{P}_N$. We say that the realization map is $(s, \alpha)$ inverse stable on $\Omega$ w.r.t. $\|\cdot\|$, if for all $\Gamma \in \Omega$ and $g \in \mathcal{R}(\Omega)$ there exists $\Phi \in \Omega$ with*

$$\mathcal{R}(\Phi) = g \quad and \quad \|\Phi - \Gamma\|_\infty \leq s\|g - \mathcal{R}(\Gamma)\|^\alpha. \tag{5}$$

In Section 2 we will see why inverse stability fails w.r.t. the uniform norm. Therefore, we consider a norm which takes into account not only the maximum error of the function values but also of the gradients. In mathematical terms, we make use of the Sobolev norm $\|\cdot\|_{W^{1,\infty}(U)}$ (on some domain $U \subseteq \mathbb{R}^d$) defined for every (locally) Lipschitz continuous function $g \colon \mathbb{R}^d \to \mathbb{R}^D$ by $\|g\|_{W^{1,\infty}(U)} := \max\{\|g\|_{L^\infty(U)}, |g|_{W^{1,\infty}(U)}\}$ with the Sobolev semi-norm $|\cdot|_{W^{1,\infty}(U)}$ given by

$$|g|_{W^{1,\infty}(U)} := \|Dg\|_{L^\infty(U)} = \operatorname*{ess\,sup}_{x \in U} \|Dg(x)\|_\infty. \tag{6}$$

See [15] for further information on Sobolev norms, and [8] for further information on the derivative of ReLU networks.

## 1.2 Implications of inverse stability for neural network optimization

We proceed by demonstrating how inverse stability opens up new perspectives on the optimization problem which arises in neural network training. Specifically, consider a loss function $\mathcal{L} \colon C(\mathbb{R}^d, \mathbb{R}^D) \to [0, \infty)$ on the space of continuous functions. For illustration, we take the commonly used mean squared error (MSE) which, for training data $((x^i, y^i))_{i=1}^n \in (\mathbb{R}^d \times \mathbb{R}^D)^n$, is given by

$$\mathcal{L}(g) = \tfrac{1}{n} \sum_{i=1}^n \|g(x^i) - y^i\|_2^2, \quad \text{for } g \in C(\mathbb{R}^d, \mathbb{R}^D). \tag{7}$$

Typically, the optimization problem is solved over some subspace of parametrizations $\Omega \subseteq \mathcal{P}_N$, i.e.

$$\min_{\Gamma \in \Omega} \mathcal{L}(\mathcal{R}(\Gamma)) = \min_{\Gamma \in \Omega} \tfrac{1}{n} \sum_{i=1}^n \|\mathcal{R}(\Gamma)(x^i) - y^i\|_2^2. \tag{8}$$

From an abstract point of view, by writing $g = \mathcal{R}(\Gamma) \in \mathcal{R}(\Omega)$, this is equivalent to the corresponding optimization problem over the space of realizations $\mathcal{R}(\Omega)$, i.e.

$$\min_{g \in \mathcal{R}(\Omega)} \mathcal{L}(g) = \min_{g \in \mathcal{R}(\Omega)} \tfrac{1}{n} \sum_{i=1}^n \|g(x^i) - y^i\|_2^2. \tag{9}$$

However, the loss landscape of the optimization problem (8) is only properly connected to the loss landscape of the optimization problem (9) if the realization map is inverse stable on $\Omega$. Otherwise a realization $g \in \mathcal{R}(\mathcal{P}_N)$ can be arbitrarily close to a global minimum in the realization space but every parametrization $\Phi$ with $\mathcal{R}(\Phi) = g$ is far away from the corresponding global minimum in the parametrization space. Moreover, local minima of (8) in the parametrization space must correspond to local minima of (9) in the realization space if and only if we have inverse stability.

**Proposition 1.2** (Parametrization minimum $\Rightarrow$ realization minimum). *Let $N \in \mathcal{A}_L$, $\Omega \subseteq \mathcal{P}_N$ and let the realization map be $(s, \alpha)$ inverse stable on $\Omega$ w.r.t. $\|\cdot\|$. Let $\Gamma_* \in \Omega$ be a local minimum of $\mathcal{L} \circ \mathcal{R}$ on $\Omega$ with radius $r > 0$, i.e. for all $\Phi \in \Omega$ with $\|\Phi - \Gamma_*\|_\infty \leq r$ it holds that*

$$\mathcal{L}(\mathcal{R}(\Gamma_*)) \leq \mathcal{L}(\mathcal{R}(\Phi)). \tag{10}$$

*Then $\mathcal{R}(\Gamma_*)$ is a local minimum of $\mathcal{L}$ on $\mathcal{R}(\Omega)$ with radius $(\frac{r}{s})^{1/\alpha}$, i.e. for all $g \in \mathcal{R}(\Omega)$ with $\|g - \mathcal{R}(\Gamma_*)\| \leq (\frac{r}{s})^{1/\alpha}$ it holds that*

$$\mathcal{L}(\mathcal{R}(\Gamma_*)) \leq \mathcal{L}(g). \tag{11}$$

See Appendix A.1.2 for a proof and Example A.1 for a counterexample in the case that inverse stability is not given. Note that in (9) we consider a problem with convex loss function but non-convex feasible set, see [34, Section 3.2]. This opens up new avenues of investigation using tools from functional analysis and allows utilizing recent results [19, 34] exploring the topological properties of neural network realization spaces.

As a concrete demonstration we provide with Theorem A.2 a strong result obtained on the realization space, which estimates the quality of a local minimum based on its radius and the approximation capabilities of the chosen architecture for a class of functions $S$. Specifically let $C > 0$, let $\Lambda \colon \mathcal{B} \to [0, \infty)$ be a quasi-convex regularizer, and define

$$S := \{f \in \mathcal{B} \colon \Lambda(f) \leq C\}. \tag{12}$$

We denote the sets of regularized parametrizations by

$$\Omega_N := \{\Phi \in \mathcal{P}_N \colon \Lambda(\mathcal{R}(\Phi)) \leq C\} \tag{13}$$

and assume that the loss function $\mathcal{L}$ is convex and $c$-Lipschitz continuous on $S$. Note that virtually all relevant loss functions are convex and locally Lipschitz continuous on $C(\mathbb{R}^d, \mathbb{R}^D)$. Employing Proposition 1.2, inverse stability can then be used to derive the following result for the practically relevant parametrized problem, showing that for sufficiently large architectures local minima of a regularized neural network optimization problem are almost optimal.

**Theorem 1.3** (Almost optimality of local parameter minima). *Assume that $S$ is compact in the $\|\cdot\|$-closure of $\mathcal{R}(\mathcal{P})$ and that for every $N \in \mathcal{A}_L$ the realization map is $(s, \alpha)$ inverse stable on $\Omega_N$ w.r.t. $\|\cdot\|$. Then for all $\varepsilon, r > 0$ there exists $n(\varepsilon, r) \in \mathcal{A}_L$ such that for every $N \in \mathcal{A}_L$ with $N_1 \geq n_1(\varepsilon, r), \ldots, N_{L-1} \geq n_{L-1}(\varepsilon, r)$ the following holds:*
*Every local minimum $\Gamma_*$ with radius at least $r$ of $\min_{\Gamma \in \Omega_N} \mathcal{L}(\mathcal{R}(\Gamma))$ satisfies*

$$\mathcal{L}(\mathcal{R}(\Gamma_*)) \leq \min_{\Gamma \in \Omega_N} \mathcal{L}(\mathcal{R}(\Gamma)) + \varepsilon. \tag{14}$$

See Appendix A.1.2 for a proof and note that here it is important to have an inverse stability result, where the parameters $(s, \alpha)$ do not depend on the size of the architecture, which we achieve for $L = 2$ and $\mathcal{B} = W^{1,\infty}$. Suitable $\Lambda$ would be Besov norms which constitute a common regularizer in image and signal processing. Moreover, note that the required size of the architecture in Theorem 1.3 can be quantified, if one has approximation rates for $S$. In particular, this approach allows the use of approximation results in order to explain the success of neural network optimization and enables a combined study of these two aspects, which, to the best of our knowledge, has not been done before. Unlike in recent literature, our result needs no assumptions on the sample set (incorporated in the loss function, see (7)), in particular we do not require "overparametrization" with respect to the sample size. Here the required size of the architecture only depends on the complexity of $S$, i.e. the class of functions one wants to approximate, the radius of the local minima of interest, the Lipschitz constant of the loss function, and the parameters of the inverse stability.

In the following we restrict ourselves to two-layer ReLU networks without biases, where we present a proof for $(4, 1/2)$ inverse stability w.r.t. the Sobolev semi-norm on a suitably regularized space of parametrizations. Both the regularizations as well as the stronger norm (compared to the uniform norm) will shown to be necessary in Section 2. We now present, in an informal way, a collection of our main results. A short proof making the connection to the formal results can be found in Appendix A.1.2.

**Corollary 1.4** (Inverse stability and implications - colloquial). *Suppose we are given data $((x^i, y^i))_{i=1}^n \in (\mathbb{R}^d \times \mathbb{R}^D)^n$ and want to solve a typical minimization problem for ReLU networks with shallow architecture $N = (d, N_1, D)$, i.e.*

$$\min_{\Gamma \in \mathcal{P}_N} \frac{1}{n} \sum_{i=1}^n \|\mathcal{R}(\Gamma)(x^i) - y^i\|_2^2. \tag{15}$$

*First we augment the architecture to $\tilde{N} = (d+2, N_1+1, D)$, while omitting the biases, and augment the samples to $\tilde{x}^i = (x_1^i, \ldots, x_d^i, 1, -1)$. Moreover, we assume that the parametrizations*

$$\Phi = \big(([a_1|\ldots|a_{N_1+1}]^T, 0), ([c_1|\ldots|c_{N_1+1}], 0)\big) \in \Omega \subseteq \mathcal{P}_{\tilde{N}} \tag{16}$$

*are regularized such that*

&emsp; *C.1 the network is balanced, i.e. $\|a_i\|_\infty = \|c_i\|_\infty$,*

&emsp; *C.2 no non-zero weight vectors in the first layer are redundant, i.e. $a_i \not\parallel a_j$,*

&emsp; *C.3 the last two coordinates of each weight vector $a_i$ are strictly positive.*

*Then for the new minimization problem*

$$\min_{\Phi \in \Omega} \frac{1}{n} \sum_{i=1}^n \|\mathcal{R}(\Phi)(\tilde{x}^i) - y^i\|_2^2 \tag{17}$$

*the following holds:*

&emsp; *1. If $\Phi_*$ is a local minimum of (17) with radius $r$, then $\mathcal{R}(\Phi_*)$ is a local minimum of $\min_{g \in \mathcal{R}(\Omega)} \frac{1}{n} \sum_{i=1}^n \|g(\tilde{x}^i) - y^i\|_2^2$ with radius at least $\frac{r^2}{16}$ w.r.t. $|\cdot|_{W^{1,\infty}}$.*

&emsp; *2. The global minimum of (17) is at least as good as the global minimum of (15), i.e.*

$$\min_{\Phi \in \Omega} \frac{1}{n} \sum_{i=1}^n \|\mathcal{R}(\Phi)(\tilde{x}^i) - y^i\|_2^2 \leq \min_{\Gamma \in \mathcal{P}_N} \frac{1}{n} \sum_{i=1}^n \|\mathcal{R}(\Gamma)(x^i) - y^i\|_2^2. \tag{18}$$

&emsp; *3. By further regularizing (17) in the sense of Theorem 1.3, we can estimate the quality of its local minima.*

This argument is not limited to the MSE loss function but works for any loss function based on evaluating the realization. The omission of bias weights is standard in neural network optimization literature [11, 13, 22, 24]. While this severely limits the functions that can be realized with a given architecture, it is sufficient to augment the problem by one dimension in order to recover the full range of functions that can be learned [1]. Here we augment by two dimensions, so that the third regularization condition C.3 can be fulfilled without loosing range. Moreover, note that, for simplicity of presentation, the regularization assumptions stated above are stricter than necessary and possible relaxations are discussed in Section 3.

## 2   Obstacles to inverse stability - degeneracies of ReLU parametrizations

In the remainder of this paper we focus on shallow ReLU networks without biases and define the corresponding space of parametrizations with architecture $N = (d, m, D)$ as $\mathcal{N}_N := \mathbb{R}^{m \times d} \times \mathbb{R}^{D \times m}$. The realization map[2] $\mathcal{R}$ is, for every $\Theta = (A, C) = \big([a_1|\ldots|a_m]^T, [c_1|\ldots|c_m]\big) \in \mathcal{N}_N$, given by

$$\mathbb{R}^d \ni x \mapsto \mathcal{R}(\Theta)(x) = C\rho(Ax) = \sum_{i=1}^m c_i \rho(\langle a_i, x \rangle). \tag{19}$$

Note that each function $x \mapsto c_i \rho(\langle a_i, x \rangle)$ represents a so-called ridge function which is zero on the half-space $\{x \in \mathbb{R}^d : \langle a_i, x \rangle \leq 0\}$ and linear with constant derivative $c_i a_i^T \in \mathbb{R}^D \times \mathbb{R}^d$ on the other half-space. Thus, the $a_i$ are the normal vectors of the separating hyperplanes $\{x \in \mathbb{R}^d : \langle a_i, x \rangle = 0\}$ and consequently we refer to the weight vectors $a_i$ also as the directions of $\Theta$. Moreover, for $\Theta \in \mathcal{N}_N$ it holds that $\mathcal{R}(\Theta)(0) = 0$ and, as long as the domain of interest $U \subseteq \mathbb{R}^d$ contains the origin, the Sobolev norm $\|\cdot\|_{W^{1,\infty}(U)}$ is equivalent to its semi-norm, since

$$\|\mathcal{R}(\Theta)\|_{L^\infty(U)} \leq \sqrt{d} \, \mathrm{diam}(U)|\mathcal{R}(\Theta)|_{W^{1,\infty}}, \tag{20}$$

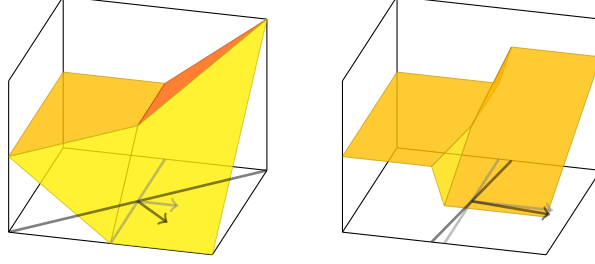

Figure 1: The figure shows $g_k$ for $k = 1, 2$.

see also inequalities of Poincaré-Friedrichs type [14, Subsection 5.8.1]. Therefore, in the rest of the paper we will only consider the Sobolev semi-norm[3]

$$|\mathcal{R}(\Theta)|_{W^{1,\infty}(U)} = \operatorname*{ess\,sup}_{x \in U} \Big\| \sum_{i \in [m]: \langle a_i, x \rangle > 0} c_i a_i^T \Big\|_\infty. \tag{21}$$

In (21) one can see that in our setting $|\cdot|_{W^{1,\infty}(U)}$ is independent of $U$ (as long as $U$ contains a neighbourhood of the origin) and will thus be abbreviated by $|\cdot|_{W^{1,\infty}}$.

## 2.1 Failure of inverse stability w.r.t. uniform norm

All proofs for this section can be found in Appendix A.2.2. We start by showing that inverse stability fails w.r.t. the uniform norm. This example is adapted from [34, Theorem 5.2] and represents, to the best of our knowledge, the only degeneracy which has already been observed before.

**Example 2.1** (Failure due to exploding gradient). *Let $\Gamma := (0,0) \in \mathcal{N}_{(2,2,1)}$ and $g_k \in \mathcal{R}(\mathcal{N}_{(2,2,1)})$ be given by (see Figure 1)*

$$g_k(x) := k\rho(\langle (k, 0), x \rangle) - k\rho(\langle (k, -\tfrac{1}{k^2}), x \rangle), \quad k \in \mathbb{N}. \tag{22}$$

*Then for every sequence $(\Phi_k)_{k \in \mathbb{N}} \subseteq \mathcal{N}_{(2,2,1)}$ with $\mathcal{R}(\Phi_k) = g_k$ it holds that*

$$\lim_{k \to \infty} \|\mathcal{R}(\Phi_k) - \mathcal{R}(\Gamma)\|_{L^\infty((-1,1)^2)} = 0 \quad and \quad \lim_{k \to \infty} \|\Phi_k - \Gamma\|_\infty = \infty. \tag{23}$$

In particular, note that inverse stability fails here even for a non-degenerate parametrization of the zero function $\Gamma = (0,0)$. However, for this type of counterexample the magnitude of the gradient of $\mathcal{R}(\Phi_k)$ needs to go to infinity, which is our motivation for looking at inverse stability w.r.t. $|\cdot|_{W^{1,\infty}}$.

## 2.2 Failure of inverse stability w.r.t. Sobolev norm

In this section we present four degenerate cases where inverse stability fails w.r.t. $|\cdot|_{W^{1,\infty}}$. This collection of counterexamples is complete in the sense that we can establish inverse stability under assumptions which are designed to exclude these four pathologies.

**Example 2.2** (Failure due to complete unbalancedness). *Let $r > 0$, $\Gamma := \big((r,0),0\big) \in \mathcal{N}_{(2,1,1)}$ and $g_k \in \mathcal{R}(\mathcal{N}_{(2,1,1)})$ be given by (see Figure 2)*

$$g_k(x) = \tfrac{1}{k}\rho(\langle (0,1), x \rangle), \quad k \in \mathbb{N}. \tag{24}$$

*Then for every $k \in \mathbb{N}$ and $\Phi_k \in \mathcal{N}_{(2,1,1)}$ with $\mathcal{R}(\Phi_k) = g_k$ it holds that*

$$|\mathcal{R}(\Phi_k) - \mathcal{R}(\Gamma)|_{W^{1,\infty}} = \tfrac{1}{k} \quad and \quad \|\Phi_k - \Gamma\|_\infty \geq r. \tag{25}$$

This is a very simple example of a degenerate parametrization of the zero function, since $\mathcal{R}(\Gamma) = 0$ regardless of choice of $r$. The issue here is that we can have a weight pair, i.e. $((r,0),0)$, where the product is independent of the value of one of the parameters. Note that in Example A.4 one can see a slightly more subtle version of this pathology by considering $\Gamma_k := \big((k,0), \tfrac{1}{k^2}\big) \in \mathcal{N}_{(2,1,1)}$ instead. In that case one could still get an inverse stability estimate for each fixed $k$; the parameters of inverse

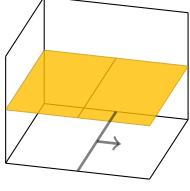 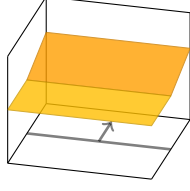 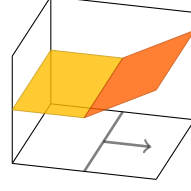 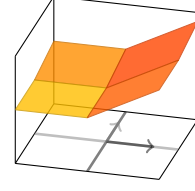

Figure 2: Shows $\mathcal{R}(\Gamma)$ ($r = 0.5$) and $g_3$.      Figure 3: Shows $\mathcal{R}(\Gamma)$ and $g_2$.

stability $(s, \alpha)$ would however deteriorate with increasing $k$. In particular this demonstrates the need for some sort of balancedness of the parametrization, i.e. control over $\|c_i\|_\infty$ and $\|a_i\|_\infty$ individually relative to $\|c_i\|_\infty \|a_i\|_\infty$.

Inverse stability is also prevented by redundant directions as the following example illustrates.

**Example 2.3** (Failure due to redundant directions). *Let*

$$\Gamma := \left( \begin{bmatrix} 1 & 0 \\ 1 & 0 \end{bmatrix}, (1,1) \right) \in \mathcal{N}_{(2,2,1)} \tag{26}$$

*and $g_k \in \mathcal{R}(\mathcal{N}_{(2,2,1)})$ be given by (see Figure 3)*

$$g_k(x) := 2\rho(\langle(1,0), x\rangle) + \tfrac{1}{k}\rho(\langle(0,1), x\rangle), \quad k \in \mathbb{N}. \tag{27}$$

*Then for every $k \in \mathbb{N}$ and $\Phi_k \in \mathcal{N}_{(2,2,1)}$ with $\mathcal{R}(\Phi_k) = g_k$ it holds that*

$$|\mathcal{R}(\Phi_k) - \mathcal{R}(\Gamma)|_{W^{1,\infty}} = \tfrac{1}{k} \quad \text{and} \quad \|\Phi_k - \Gamma\|_\infty \geq 1. \tag{28}$$

The next example shows that not only redundant weight vectors can cause issues, but also weight vectors of opposite direction, as they would allow for a (balanced) degenerate parametrization of the zero function.

**Example 2.4** (Failure due to opposite weight vectors 1). *Let $a_i \in \mathbb{R}^d$, $i \in [m]$, be pairwise linearly independent with $\|a_i\|_\infty = 1$ and $\sum_{i=1}^m a_i = 0$. We define*

$$\Gamma := \left( [a_1| \ldots |a_m| - a_1| \ldots | - a_m]^T, (1, \ldots, 1, -1, \ldots, -1) \right) \in \mathcal{N}_{(d,2m,1)}. \tag{29}$$

*Now let $v \in \mathbb{R}^d$ with $\|v\|_\infty = 1$ be linearly independent to each $a_i$, $i \in [m]$, and let $g_k \in \mathcal{R}(\mathcal{N}_{(d,2m,1)})$ be given by (see Figure 4)*

$$g_k(x) = \tfrac{1}{k}\rho(\langle v, x\rangle), \quad k \in \mathbb{N}. \tag{30}$$

*Then there exists a constant $C > 0$ such that for every $k \in \mathbb{N}$ and every $\Phi_k \in \mathcal{N}_{(d,2m,1)}$ with $\mathcal{R}(\Phi_k) = g_k$ it holds that*

$$|\mathcal{R}(\Phi_k) - \mathcal{R}(\Gamma)|_{W^{1,\infty}} = \tfrac{1}{k} \quad \text{and} \quad \|\Phi_k - \Gamma\|_\infty \geq C. \tag{31}$$

Thus we will need an assumption which prevents each individual $\Gamma$ in our restricted set from having pairwise linearly dependent weight vectors, i.e. coinciding hyperplanes of non-differentiability. This, however, does not suffice as is demonstrated by the next example, which shows that the relation between the hyperplanes of the two realizations matters.

**Example 2.5** (Failure due to opposite weight vectors 2). *We define the weight vectors*

$$a_1^k = (k, k, \tfrac{1}{k}), \quad a_2^k = (-k, k, \tfrac{1}{k}), \quad a_3^k = (0, -\sqrt{2}k, \tfrac{1}{\sqrt{2}k}), \quad c^k = (k, k, \sqrt{2}k) \tag{32}$$

*and consider the parametrizations (see Figure 5)*

$$\Gamma_k := \left( \left[ -a_1^k| - a_2^k| - a_3^k \right]^T, c^k \right) \in \mathcal{N}_{(3,3,1)}, \quad \Theta_k := \left( [a_1^k|a_2^k|a_3^k]^T, c^k \right) \in \mathcal{N}_{(3,3,1)}. \tag{33}$$

*Then for every $k \in \mathbb{N}$ and every $\Phi_k \in \mathcal{N}_{(3,3,1)}$ with $\mathcal{R}(\Phi_k) = \mathcal{R}(\Theta_k)$ it holds that*

$$|\mathcal{R}(\Phi_k) - \mathcal{R}(\Gamma_k)|_{W^{1,\infty}} = 3 \quad \text{and} \quad \|\Phi_k - \Gamma_k\|_\infty \geq k. \tag{34}$$

Note that $\Gamma$ and $\Theta$ need to have multiple exactly opposite weight vectors which add to something small (compared to the size of the individual vectors), but not zero, since otherwise reparametrization would be possible (see Lemma A.5).

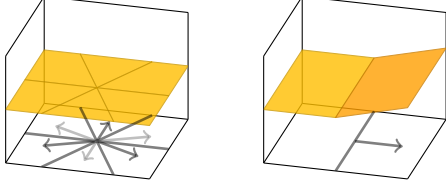
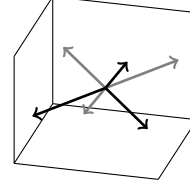

Figure 4: Shows $\mathcal{R}(\Gamma)$ and $g_3$ ($a_1 = (1, -\frac{1}{2})$, $a_2 = (-1, -\frac{1}{2})$, $a_3 = (0, 1)$, $v = (1, 0)$).

Figure 5: Shows the weight vectors of $\Theta_2$ (grey) and $\Gamma_2$ (black).

## 3   Inverse stability for two-layer ReLU Networks

We now establish an inverse stability result using assumptions designed to exclude the pathologies from the previous section. First we present a rather technical theorem for output dimension one which considers a parametrization $\Gamma$ in the unrestricted parametrization space $\mathcal{N}_N$ and a function $g$ in the the corresponding function space $\mathcal{R}(\mathcal{N}_N)$. The aim is to use assumptions which are as weak as possible, while allowing us to find a parametrization $\Phi$ of $g$, whose distance to $\Gamma$ can be bounded relative to $|g - \mathcal{R}(\Gamma)|_{W^{1,\infty}}$. We then continue by defining a restricted parametrization space $\mathcal{N}_N^*$, for which we get uniform inverse stability (meaning that we get the same estimate for every $\Gamma \in \mathcal{N}_N^*$).

**Theorem 3.1** (Inverse stability at $\Gamma \in \mathcal{N}_N$). *Let $d, m \in \mathbb{N}$, $N := (d, m, 1)$, $\beta \in [0, \infty)$, let $\Gamma = \left( \left[ a_1^\Gamma \mid \ldots \mid a_m^\Gamma \right]^T, c^\Gamma \right) \in \mathcal{N}_N$, $g \in \mathcal{R}(\mathcal{N}_N)$, and let $I^\Gamma := \{i \in [m] \colon a_i^\Gamma \neq 0\}$. Assume that the following conditions are satisfied:*

*C.1  It holds for all $i \in [m]$ with $\|c_i^\Gamma a_i^\Gamma\|_\infty \leq 2|g - \mathcal{R}(\Gamma)|_{W^{1,\infty}}$ that $|c_i^\Gamma|, \|a_i^\Gamma\|_\infty \leq \beta$.*

*C.2  It holds for all $i, j \in I^\Gamma$ with $i \neq j$ that $\frac{a_j^\Gamma}{\|a_j^\Gamma\|_\infty} \neq \frac{a_i^\Gamma}{\|a_i^\Gamma\|_\infty}$.*

*C.3  There exists a parametrization $\Theta = \left( \left[ a_1^\Theta \mid \ldots \mid a_m^\Theta \right]^T, c^\Theta \right) \in \mathcal{N}_N$ such that $\mathcal{R}(\Theta) = g$ and*

*(a)  it holds for all $i, j \in I^\Gamma$ with $i \neq j$ that $\frac{a_j^\Gamma}{\|a_j^\Gamma\|_\infty} \neq -\frac{a_i^\Gamma}{\|a_i^\Gamma\|_\infty}$ and for all $i, j \in I^\Theta$ with $i \neq j$ that $\frac{a_j^\Theta}{\|a_j^\Theta\|_\infty} \neq -\frac{a_i^\Theta}{\|a_i^\Theta\|_\infty}$,*

*(b)  it holds for all $i \in I^\Gamma$, $j \in I^\Theta$ that $\frac{a_i^\Gamma}{\|a_i^\Gamma\|_\infty} \neq -\frac{a_j^\Theta}{\|a_j^\Theta\|_\infty}$*

*where $I^\Theta := \{i \in [m] \colon a_i^\Theta \neq 0\}$.*

*Then there exists a parametrization $\Phi \in \mathcal{N}_N$ with*

$$\mathcal{R}(\Phi) = g \quad and \quad \|\Phi - \Gamma\|_\infty \leq \beta + 2|g - \mathcal{R}(\Gamma)|_{W^{1,\infty}}^{\frac{1}{2}}. \tag{35}$$

The proof can be found in Appendix A.3.2. Note that each of the conditions in the theorem above corresponds directly to one of the pathologies in Section 2.2. Condition C.1, which deals with unbalancedness, only imposes an restriction on the weight pairs whose product is small compared to the distance of $\mathcal{R}(\Gamma)$ and $g$. As can be guessed from Example 2.2 and seen in the proof of Theorem 3.1, such a balancedness assumption is in fact only needed to deal with degenerate cases, where $\mathcal{R}(\Gamma)$ and $g$ have parts with mismatching directions of negligible magnitude. Otherwise a matching reparametrization is always possible. Note that a balanced $\Gamma$ (i.e. $|c_i^\Gamma| = \|a_i^\Gamma\|_\infty$) satisfies Condition C.1 with $\beta = (2|g - \mathcal{R}(\Gamma)|_{W^{1,\infty}})^{1/2}$.

It is also possible to relax the balancedness assumption by only requiring $|c_i^\Gamma|$ and $\|\Gamma_i\|_\infty$ to be close to $\|c_i^\Gamma a_i^\Gamma\|_\infty^{1/2}$, which would still give a similar estimate but with a worse exponent. In order to see that requiring balancedness does not restrict the space of realizations, observe that the ReLU is positively homogeneous (i.e. $\rho(\lambda x) = \lambda \rho(x)$ for all $\lambda \geq 0$, $x \in \mathbb{R}$). Thus balancedness can always be achieved simply by rescaling.

Condition C.2 requires $\Gamma$ to have no redundant directions, the necessity of which is demonstrated by Example 2.3. Note that prohibiting redundant directions does not restrict the space of realizations,

see (87) in the appendix for details. From a practical point of view, enforcing this condition could be achieved by a regularization term using a barrier function. Alternatively on could employ a non-standard approach of combining such redundant neurons by changing one of them according to (87) and either setting the other one to zero or removing it entirely[4].

From a theoretical perspective the first two conditions are rather mild, in the sense that they only restrict the space of parametrizations and not the corresponding space of realizations. Specifically we can define the restricted parametrization space

$$\mathcal{N}'_{(d,m,D)} := \{\Gamma \in \mathcal{N}_{(d,m,D)} \colon \|c_i^\Gamma\|_\infty = \|a_i^\Gamma\|_\infty \text{ for all } i \in [m] \text{ and } \Gamma \text{ satisfies C.2}\} \qquad (36)$$

for which we have $\mathcal{R}(\mathcal{N}'_N) = \mathcal{R}(\mathcal{N}_N)$. Note that the above definition as well as the following definition and theorem are for networks with arbitrary output dimensions, as the balancedness condition makes this extension rather straightforward.

In order to satisfy Conditions C.3a and C.3b we need to restrict the parametrization space in a way which also restricts the corresponding space of realizations. One possibility to do so is the following approach, which also incorporates the previous restrictions as well as the transition to networks without biases.

**Definition 3.2** (Restricted parametrization space). *Let* $N = (d, m, D) \in \mathbb{N}^3$. *We define*

$$\mathcal{N}_N^* := \left\{\Gamma \in \mathcal{N}'_N \colon (a_i^\Gamma)_{d-1}, (a_i^\Gamma)_d > 0 \text{ for all } i \in [m]\right\}. \qquad (37)$$

While we no longer have $\mathcal{R}(\mathcal{N}_N^*) = \mathcal{R}(\mathcal{N}_N)$, Lemma A.6 shows that for every $\Theta \in \mathcal{P}_{(d,m,D)}$ there exists $\Gamma \in \mathcal{N}_{(d+2,m+1,D)}^*$ such that for all $x \in \mathbb{R}^d$ it holds that

$$\mathcal{R}(\Gamma)(x_1, \ldots, x_d, 1, -1) = \mathcal{R}(\Theta)(x_1, \ldots, x_d). \qquad (38)$$

In particular, this means that for any optimization problem over an unrestricted parametrization space $\mathcal{P}_{(d,m,D)}$, there is a corresponding optimization problem over the parametrization space $\mathcal{N}_{(d+2,m+1,D)}^*$ whose solution is at least as good (see Corollary 1.4). Our main result now states that for such a restricted parametrization space we have uniform $(4, 1/2)$ inverse stability w.r.t. $|\cdot|_{W^{1,\infty}}$, a proof of which can be found in Appendix A.3.2.

**Theorem 3.3** (Inverse stability on $\mathcal{N}_N^*$). *Let* $N \in \mathbb{N}^3$. *For all* $\Gamma \in \mathcal{N}_N^*$ *and* $g \in \mathcal{R}(\mathcal{N}_N^*)$ *there exists a parametrization* $\Phi \in \mathcal{N}_N^*$ *with*

$$\mathcal{R}(\Phi) = g \quad \text{and} \quad \|\Phi - \Gamma\|_\infty \leq 4|g - \mathcal{R}(\Gamma)|_{W^{1,\infty}}^{\frac{1}{2}}. \qquad (39)$$

# 4 Outlook

This contribution investigates the potential insights which may be gained from studying the optimization problem over the space of realizations, as well as the difficulties encountered when trying to connect it to the parametrized problem. While Theorem 1.3 and Theorem 3.3 offer some compelling preliminary answers, there are multiple ways in which they can be extended.

To obtain our inverse stability result for shallow ReLU networks we studied sums of ridge functions. Extending this result to deep ReLU networks requires understanding their behaviour under composition. In particular, we have ridge functions which vanish on some half space, i.e. colloquially speaking each neuron may "discard half the information" it receives from the previous layer. This introduces a new type of degeneracy, which one will have to deal with.

Another interesting direction is an extension to inverse stability w.r.t. some weaker norm like $\|\cdot\|_{L^\infty}$ or a fractional Sobolev norm under stronger restrictions on the space of parametrizations (see Lemma A.7 for a simple approach using very strong restrictions).

Lastly, note that Theorem 1.3 is not specific to the ReLU activation function and thus also incentivizes the study of inverse stability for any other activation function.

From an applied point of view, Conditions C.1-C.3 motivate the implementation of corresponding regularization (i.e. penalizing unbalancedness and redundancy in the sense of parallel weight vectors) in state-of-the-art networks, in order to explore whether preventing inverse stability leads to improved performance in practice. Note that there already are results using, e.g. *cosine similarity*, as regularizer to prevent parallel weight vectors [4, 35] as well as approaches, called *Sobolev Training*, reporting better generalization and data-efficiency by employing a Sobolev norm based loss [12].

## Acknowledgment

The research of JB and DE was supported by the Austrian Science Fund (FWF) under grants I3403-N32 and P 30148. The authors would like to thank Pavol Harár for helpful comments.

## Footnotes

[1]On the finite dimensional vector space $\mathcal{P}_N$ all norms are equivalent and we take w.l.o.g. the maximum norm $\|\Theta\|_\infty$, i.e. the maximum of the absolute values of the entries of the $A_\ell$ and $b_\ell$.

[2]This is a slight abuse of notation, justified by the the fact that $\mathcal{R}$ acts the same on $\mathcal{P}_N$ with zero biases $b_1, b_2$ and weights $A_1 = A$ and $A_2 = C$.

[3]For $m \in \mathbb{N}$ we abbreviate $[m] := \{1, \dots, m\}$.

[4]This could be of interest in the design of dynamic network architectures [26, 28, 40] and is also closely related to the co-adaption of neurons, to counteract which, dropout was invented [21].

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
