[Supplementary Material · supp.pdf]

# A Appendix - Proofs and Additional Material

## A.1 Section 1

### A.1.1 Additional Material

**Example A.1** (Without inverse stability: parameter minimum $\not\Rightarrow$ realization minimum). *Consider the two domains*

$$D_1 := \{(x_1, x_2) \in (-1,1)^2 : x_2 > |x_1|\}, \quad D_2 := \{(x_1, x_2) \in (-1,1)^2 : x_1 > |x_2|\}. \quad (40)$$

*For simplicity of presentation, assume we are given two samples $x^1 \in D_1$, $x^2 \in D_2$ with labels $y^1 = 0$, $y^2 = 1$. The corresponding MSE is*

$$\mathcal{L}(g) = \tfrac{1}{2}\big((g(x^1))^2 + (g(x^2) - 1)^2\big) \quad (41)$$

*for every $g \in C(\mathbb{R}^2, \mathbb{R})$. Let the zero realization be parametrized by[5]*

$$\Gamma_* = (0, (-1, 0)) \in \mathcal{N}_{(2,1,1)} \quad (42)$$

*with loss $\mathcal{L}(\mathcal{R}(\Gamma_*)) = \tfrac{1}{2}$. Note that changing each weight by less than $\tfrac{1}{2}$ does not decrease the loss, as this rotates the vector $(-1, 0)$ by at most $45°$. Thus $\Gamma_*$ is a local minimum in the parametrization space. However, the sequence of realizations given by*

$$g_k(x) = \tfrac{1}{k}\rho(x_1 - x_2) = \mathcal{R}((1, -1), \tfrac{1}{k}) \quad (43)$$

*satisfies that*

$$\|g_k - \mathcal{R}(\Gamma_*)\|_{W^{1,\infty}((-1,1)^2)} = \|g_k\|_{W^{1,\infty}((-1,1)^2)} \leq \tfrac{1}{k} \quad (44)$$

*and*

$$\mathcal{L}(g_k) = \tfrac{1}{2}(g_k(x^2) - 1)^2 < \tfrac{1}{2} = \mathcal{L}(\mathcal{R}(\Gamma_*)), \quad (45)$$

*see Figure 6. Accordingly, $\mathcal{R}(\Gamma_*)$ is not a local minimum in the realization space even w.r.t. the Sobolev norm. The problem occurs, since inverse stability fails due to unbalancedness of $\Gamma_*$.*

Figure 6: The figure shows the samples $((x^i, y^i))_{i=1,2}$, the realization $\mathcal{R}(\Gamma_*)$ of the local parameter minimum (left) and $g_3$ (right).

**Theorem A.2** (Quality of local realization minima). *Assume that*

$$\sup_{f \in S} \; \inf_{\Phi \in \Omega_N} \|\mathcal{R}(\Phi) - f\| < \eta \quad \text{(approximability)}. \quad (46)$$

*Let $g_*$ be a local minimum with radius $r' \geq 2\eta$ of the optimization problem $\min_{g \in \mathcal{R}(\Omega_N)} \mathcal{L}(g)$. Then it holds for every $g \in \mathcal{R}(\Omega_N)$ (in particular for every global minimizer) that*

$$\mathcal{L}(g_*) \leq \mathcal{L}(g) + \tfrac{2c}{r'}\|g_* - g\|\eta. \quad (47)$$

*Proof.* Define $\lambda := \frac{r'}{2\|g - g_*\|}$ and $f := (1 - \lambda)g_* + \lambda g \in S$. Due to (46) there is $\Phi \in \Omega_N$ such that $\|\mathcal{R}(\Phi) - f\| \leq \eta$ and by the assumptions on $g_*$ and $\mathcal{L}$ it holds that

$$\mathcal{L}(g_*) \leq \mathcal{L}(\mathcal{R}(\Phi)) \leq \mathcal{L}(f) + c\eta \leq (1 - \lambda)\mathcal{L}(g_*) + \lambda\mathcal{L}(g) + c\eta.$$

This completes the proof. See Figure 7 for illustration. $\qquad \square$

Figure 7: The figure illustrates the proof idea of Theorem A.2. Note that decreasing $\eta$, $c$, $\|g_* - g\|$ or increasing $r'$ leads to a better local minimum due to the convexity of the loss function (red).

### A.1.2 Proofs

*Proof of Proposition 1.2.* By Definition 1.1 we know that for every $g \in \mathcal{R}(\Omega)$ with $\|g - \mathcal{R}(\Gamma_*)\| \leq (\frac{r}{s})^{1/\alpha}$ there exists $\Phi \in \Omega$ with

$$\mathcal{R}(\Phi) = g \quad \text{and} \quad \|\Phi - \Gamma_*\|_\infty \leq s\|g - \mathcal{R}(\Gamma_*)\|^\alpha \leq r. \tag{48}$$

Therefore by assumption it holds that

$$\mathcal{L}(\mathcal{R}(\Gamma_*)) \leq \mathcal{L}(\mathcal{R}(\Phi)) = \mathcal{L}(g). \tag{49}$$

which proves the claim. $\qquad\square$

*Proof of Theorem 1.3.* Let $\varepsilon, r > 0$, define $r' := (\frac{r}{s})^{1/\alpha}$ and $\eta := \min\{(\frac{2c}{r'}\operatorname{diam}(S))^{-1}\varepsilon, \frac{r'}{2}\}$. Then compactness of $S$ implies the existence of an architecture $n(\varepsilon, r) \in \mathcal{A}_L$ such that for every $N \in \mathcal{A}_L$ with $N_1 \geq n_1(\varepsilon, r), \ldots, N_{L-1} \geq n_{L-1}(\varepsilon, r)$ the approximability assumption (46) is satisfied. Let now $\Gamma_*$ be a local minimum with radius at least $r$ of $\min_{\Gamma \in \Omega_N} \mathcal{L}(\mathcal{R}(\Gamma))$. As we assume uniform $(s, \alpha)$ inverse stability, Proposition 1.2 implies that $\mathcal{R}(\Gamma_*)$ is a local minimum of the optimization problem $\min_{g \in \mathcal{R}(\Omega_N)} \mathcal{L}(g)$ with radius at least $r' = (\frac{r}{s})^{1/\alpha} \geq 2\eta$. Theorem A.2 establishes the claim. $\qquad\square$

*Proof of Corollary 1.4.* We simply combine the main observations from our paper. First, note that the assumptions imply that the restricted parametrization space $\Omega$, which we are optimizing over, is the space $\mathcal{N}^*_{(d+2, N_1+1, D)}$ from Definition 3.2. Secondly, Theorem 3.3 implies that the realization map is $(4, 1/2)$ inverse stable on $\Omega$. Thus, Proposition 1.2 directly proves Claim 1. For the proof of Claim 2 we make use of Lemma A.6. It implies that for every $\Theta \in \mathcal{P}_{(d, N_1, D)}$ there exists $\Gamma \in \Omega$ such that it holds that

$$\frac{1}{n} \sum_{i=1}^n \|\mathcal{R}(\Gamma)(\tilde{x}^i) - y^i\|^2 = \frac{1}{n} \sum_{i=1}^n \|\mathcal{R}(\Theta)(x^i) - y^i\|^2, \tag{50}$$

which proves the claim. $\qquad\square$

### A.2 Section 2

#### A.2.1 Additional Material

**Lemma A.3** (Reparametrization in case of linearly independent weight vectors)**.** *Let*

$$\Theta = (A^\Theta, C^\Theta) = \left([a_1^\Theta | \ldots | a_m^\Theta]^T, [c_1^\Theta | \ldots | c_m^\Theta]\right) \in \mathcal{N}_{(d, m, D)} \tag{51}$$

with linearly independent weight vectors $(a_i^\Theta)_{i=1}^m$ and $\min_{i \in [m]} \|c_i^\Theta\|_\infty > 0$ and let

$$\Phi = (A^\Phi, B^\Phi) = \left([a_1^\Phi | \ldots | a_m^\Phi]^T, [c_1^\Phi | \ldots | c_m^\Phi]\right) \in \mathcal{N}_{(d,m,D)} \tag{52}$$

with $\mathcal{R}(\Phi) = \mathcal{R}(\Theta)$. Then there exists a permutation $\pi \colon [m] \to [m]$ such that for every $i \in [m]$ there exist $\lambda_i \in (0, \infty)$ with

$$a_i^\Phi = \lambda_i a_{\pi(i)}^\Theta \quad \text{and} \quad c_i^\Phi = \tfrac{1}{\lambda_i} c_{\pi(i)}^\Theta. \tag{53}$$

*This means that, up to reordering and rebalancing, $\Theta$ is the unique parametrization of $\mathcal{R}(\Theta)$.*

*Proof.* First we define for every $s \in \{0,1\}^m$ the corresponding open orthant

$$O^s := \{x \in \mathbb{R}^m \colon x_1(2s_1 - 1) > 0, \ldots, x_m(2s_m - 1) > 0\} \subseteq \mathbb{R}^m. \tag{54}$$

By assumption $A^\Theta$ has rank $m$, i.e. is surjective, and therefore the preimages of the orthants

$$H^s := \{x \in \mathbb{R}^d \colon A^\Theta x \in O^s\} \subseteq \mathbb{R}^d, \quad s \in \{0,1\}^m, \tag{55}$$

are disjoint, non-empty open sets. Note that on each $H^s$ the realization $\mathcal{R}(\Theta)$ is linear with

$$\mathcal{R}(\Theta)(x) = C^\Theta \operatorname{diag}(s) A^\Theta x \quad \text{and} \quad D\mathcal{R}(\Theta)(x) = C^\Theta \operatorname{diag}(s) A^\Theta. \tag{56}$$

Since $A^\Theta$ has full row rank, it has a right inverse. Thus we have for $s, t \in \{0,1\}^m$ that

$$C^\Theta \operatorname{diag}(s) A^\Theta = C^\Theta \operatorname{diag}(t) A^\Theta \implies C^\Theta \operatorname{diag}(s) = C^\Theta \operatorname{diag}(t). \tag{57}$$

Note that $C^\Theta \operatorname{diag}(s) = C^\Theta \operatorname{diag}(t)$ can only hold if $s = t$ due to the assumptions that $\|c_i^\Theta\|_\infty \neq 0$ for all $i \in [m]$. Thus the above establishes that for $s, t \in \{0,1\}^m$ it holds that

$$C^\Theta \operatorname{diag}(s) A^\Theta = C^\Theta \operatorname{diag}(t) A^\Theta \quad \text{if and only if} \quad s = t, \tag{58}$$

i.e. $\mathcal{R}(\Theta)$ has different derivatives on its $2^m$ linear regions. In order for $\mathcal{R}(\Phi)$ to have matching linear regions and matching derivatives on each one of them, there must exist a permutation matrix $P \in \{0,1\}^{m \times m}$ such that for every $s \in \{0,1\}^m$

$$PA^\Phi x \in O^s \quad \text{for every } x \in H^s. \tag{59}$$

Thus, there exist $(\lambda_i)_{i=1}^m \in (0, \infty)^m$ such that

$$A^\Phi = \operatorname{diag}(\lambda_1, \ldots, \lambda_m) P^T A^\Theta. \tag{60}$$

The assumption that $D\mathcal{R}(\Theta) = D\mathcal{R}(\Psi)$, together with (56) for $s = (1, \ldots, 1)$, implies that

$$C^\Phi = C^\Theta P \operatorname{diag}(\tfrac{1}{\lambda_1}, \ldots, \tfrac{1}{\lambda_m}), \tag{61}$$

which proves the claim. $\qquad\square$

**Example A.4** (Failure due to unbalancedness)**.** *Let*

$$\Gamma_k := \left((k,0), \tfrac{1}{k^2}\right) \in \mathcal{N}_{(2,1,1)}, \quad k \in \mathbb{N}, \tag{62}$$

*and $g_k \in \mathcal{R}(\mathcal{N}_{(2,1,1)})$ be given by*

$$g_k(x) = \tfrac{1}{k}\rho(\langle(0,1), x\rangle), \quad k \in \mathbb{N}. \tag{63}$$

*The only way to parametrize $g_k$ is $g_k(x) = \mathcal{R}(\Phi_k)(x) = c\rho(\langle(0,a), x\rangle)$ with $a, c > 0$ (see Lemma A.3), and we have*

$$|\mathcal{R}(\Phi_k) - \mathcal{R}(\Gamma_k)|_{W^{1,\infty}} \leq \tfrac{1}{k} \quad \text{and} \quad \|\Phi_k - \Gamma_k\|_\infty \geq k. \tag{64}$$

**Lemma A.5.** *Let $d, m \in \mathbb{N}$ and $a_i \in \mathbb{R}^d$, $i \in [m]$, such that $\sum_{i \in [m]} a_i = 0$. Then it holds for all $x \in \mathbb{R}^d$ that*

$$\sum_{i \in [m]} \rho(\langle a_i, x\rangle) = \sum_{i \in [m]} \rho(\langle -a_i, x\rangle). \tag{65}$$

*Proof.* By assumption we have for all $x \in \mathbb{R}^d$ that $\sum_{i \in [m]} \langle a_i, x\rangle = 0$. This implies for all $x \in \mathbb{R}^d$ that

$$\sum_{i \in [m] \colon \langle a_i, x\rangle \geq 0} \langle a_i, x\rangle - \sum_{i \in [m]} \langle a_i, x\rangle = \sum_{i \in [m] \colon \langle a_i, x\rangle \leq 0} -\langle a_i, x\rangle, \tag{66}$$

which proves the claim. $\qquad\square$

### A.2.2  Proofs

*Proof of Example 2.1.* We have for every $k \in \mathbb{N}$ that

$$\|g_k\|_{L^\infty((-1,1)^2)} \leq \tfrac{1}{k} \quad \text{and} \quad |g_k|_{W^{1,\infty}} = k^2. \tag{67}$$

Assume that there exists sequence of networks $(\Phi_k)_{k \in \mathbb{N}} \subseteq \mathcal{N}_{(2,2,1)}$ with $\mathcal{R}(\Phi_k) = g_k$ and with uniformly bounded parameters, i.e. $\sup_{k \in \mathbb{N}} \|\Phi_k\|_\infty < \infty$. Note that there exists a constant $C$ (depending only on the network architecture) such that the realizations $\mathcal{R}(\Phi_k)$ are Lipschitz continuous with

$$\mathrm{Lip}(\mathcal{R}(\Phi_k)) \leq C\|\Phi_k\|_\infty^2$$

(see [34, Prop. 5.1]). It follows that $|\mathcal{R}(\Phi_k)|_{W^{1,\infty}} \leq \mathrm{Lip}(\mathcal{R}(\Phi_k))$ is uniformly bounded which contradicts (67). ∎

*Proof of Example 2.2.* The only way to parametrize $g_k$ is $g_k(x) = \mathcal{R}(\Phi_k)(x) = c\rho(\langle(0,a),x\rangle)$ with $a, c > 0$ (see also Lemma A.3), which proves the claim. ∎

*Proof of Example 2.3.* Any parametrization of $g_k$ must be of the form $\Phi_k := (A, c) \in \mathbb{R}^{2\times 2} \times \mathbb{R}^{1\times 2}$ with

$$A = \begin{bmatrix} a_1 & 0 \\ 0 & a_2 \end{bmatrix} \quad \text{or} \quad A = \begin{bmatrix} 0 & a_2 \\ a_1 & 0 \end{bmatrix} \tag{68}$$

(see Lemma A.3). Thus it holds that $\|\Phi_k - \Gamma\|_\infty \geq \|(1,0) - (0,a_2)\|_\infty \geq 1$ and the proof is completed by direct calculation. ∎

*Proof of Example 2.4.* Let $\Phi_k$ be an arbitrary parametrization of $g_k$ given by

$$\Phi_k = \left([\tilde{a}_1|\tilde{a}_2|\dots|\tilde{a}_{2m}]^T, \tilde{c}\right) \in \mathcal{N}_{(d,2m,1)} \tag{69}$$

As $g_k$ has two linear regions separated by the hyperplane with normal vector $v$, there exists $j \in [2m]$ and $\lambda \in \mathbb{R} \setminus \{0\}$ such that

$$\tilde{a}_j = \lambda v. \tag{70}$$

The distance of any weight vector $\pm a_i$ of $\Gamma$ to the line $\{\lambda v \colon \lambda \in \mathbb{R}\}$ can be lower bounded by

$$\| \pm a_i - \lambda v\|_\infty^2 \geq \tfrac{1}{d}\| \pm a_i - \lambda v\|_2^2 \geq \tfrac{1}{d^2}\left[\|a_i\|_2^2\|v\|_2^2 - \langle a_i, v\rangle^2\right], \quad i \in [m], \lambda \in \mathbb{R}. \tag{71}$$

The Cauchy-Schwarz inequality and the linear independence of $v$ to each $a_i$, $i \in [m]$, establishes that $C := \tfrac{1}{d^2}\min_{i \in [m]}\left[\|a_i\|_2^2\|v\|_2^2 - \langle a_i, v\rangle^2\right] > 0$. Together with the fact that $\mathcal{R}(\Gamma) = 0$, this completes the proof. ∎

*Proof of Example 2.5.* Since $x = \rho(x) - \rho(-x)$ for every $x \in \mathbb{R}$, the difference of the realizations is linear, i.e.

$$\mathcal{R}(\Theta_k) - \mathcal{R}(\Gamma_k) = \langle c_1^k a_1^k + c_2^k a_2^k + c_3^k a_3^k, x\rangle = \langle(0,0,3), x\rangle \tag{72}$$

and thus the difference of the gradients is constant, i.e.

$$|\mathcal{R}(\Theta_k) - \mathcal{R}(\Gamma_k)|_{W^{1,\infty}} = 3, \quad k \in \mathbb{N}. \tag{73}$$

However, regardless of the balancing and reordering of the weight vectors $a_i^k$, $i \in [3]$, we have that

$$\|\Theta_k - \Gamma_k\|_\infty \geq k. \tag{74}$$

By Lemma A.3, up to balancing and reordering, there does not exist any other parametrization of $\Theta_k$ with the same realization. ∎

### A.3 Section 3

#### A.3.1 Additional Material

**Lemma A.6.** *Let $d, m, D \in \mathbb{N}$ and $\Theta \in \mathcal{P}_{(d,m,D)}$. Then there exists $\Gamma \in \mathcal{N}^*_{(d+2,m+1,D)}$ such that for all $x \in \mathbb{R}^d$ it holds that*

$$\mathcal{R}(\Gamma)(x_1, \ldots, x_d, 1, -1) = \mathcal{R}(\Theta)(x). \tag{75}$$

*Proof.* Since $\Theta \in \mathcal{P}_{(d,m,D)}$ it can be written as

$$\Theta = \left( (A, b), (c, e) \right) = \left( ([a_1| \ldots |a_m]^T, b), ([c_1| \ldots |c_m], e) \right) \tag{76}$$

with

$$\mathcal{R}(\Theta)(x) = \sum_{i=1}^m c_i \rho(\langle a_i, x \rangle + b_i) + e, \quad x \in \mathbb{R}^d, \tag{77}$$

where $A \in \mathbb{R}^{m \times d}$, $b \in \mathbb{R}^m$, $C \in \mathbb{R}^{D \times m}$, and $e \in \mathbb{R}^D$. We define for $i \in [m]$

$$b_i^+ := \begin{cases} b_i + 1 & : b_i \geq 0 \\ 1 & : b_i < 0 \end{cases}, \quad \text{and} \quad b_i^- := \begin{cases} 1 & : b_i \geq 0 \\ -b_i + 1 & : b_i < 0 \end{cases} \tag{78}$$

and observe that $b_i^+ > 0$, $b_i^- > 0$, and $b_i^+ - b_i^- = b_i$. For $i \in [m]$ let

$$c_i^* := \begin{cases} c_i & : \|c_i\|_\infty \neq 0 \\ (1, \ldots, 1) & : \|c_i\|_\infty = 0 \end{cases} \tag{79}$$

and

$$a_i^* := \begin{cases} (a_{i,1}, \ldots, a_{i,d}, b_i^+, b_i^-) & : \|c_i\|_\infty \neq 0 \\ (0, \ldots, 0, 1, 1) & : \|c_i\|_\infty = 0 \end{cases}. \tag{80}$$

Note that we have

$$\mathcal{R}(\Theta)(x) = \sum_{i=1}^m c_i^* \rho(\langle a_i^*, (x_1, \ldots, x_d, 1, -1) \rangle) + e, \quad x \in \mathbb{R}^d. \tag{81}$$

To include the second bias $e$ let

$$c_{m+1}^* := \begin{cases} e & : e \neq 0 \\ (1, \ldots, 1) & : e = 0 \end{cases}, \quad \text{and} \quad a_{m+1}^* := \begin{cases} (0, \ldots, 0, 2, 1) & : e \neq 0 \\ (0, \ldots, 0, 1, 1) & : e = 0 \end{cases}. \tag{82}$$

In order to balance the network, let $a_i^\Gamma = a_i^* \left( \frac{\|c_i^*\|_\infty}{\|a_i^*\|_\infty} \right)^{1/2}$ and $c_i^\Gamma = c_i^* \left( \frac{\|a_i^*\|_\infty}{\|c_i^*\|_\infty} \right)^{1/2}$ for every $i \in [m+1]$. Then the claim follows by direct computation. $\qquad \square$

#### A.3.2 Proofs

*Proof of Theorem 3.1.* Without loss of generality[6], we can assume for all $i \in [m]$ that $a_i^\Theta = 0$ if and only if $c_i^\Theta = 0$. We now need to show that there always exists a way to reparametrize $\mathcal{R}(\Theta)$ such that the architecture remains the same and (35) is satisfied. For simplicity of notation we will write $r := |g - \mathcal{R}(\Gamma)|_{W^{1,\infty}}$ throughout the proof. Let $f_i^\Gamma : \mathbb{R}^d \to \mathbb{R}$ resp. $f_i^\Theta : \mathbb{R}^d \to \mathbb{R}$ be the part that is contributed by the $i$-th neuron, i.e.

$$\mathcal{R}(\Gamma) = \sum_{i=1}^m f_i^\Gamma \quad \text{with} \quad f_i^\Gamma(x) := c_i^\Gamma \rho(\langle a_i^\Gamma, x \rangle), \tag{83}$$

$$g = \mathcal{R}(\Theta) = \sum_{i=1}^m f_i^\Theta \quad \text{with} \quad f_i^\Theta(x) := c_i^\Theta \rho(\langle a_i^\Theta, x \rangle). \tag{84}$$

Further let

$$
\begin{aligned}
H_{\Gamma,i}^+ &:= \{x \in \mathbb{R}^d \colon \langle a_i^\Gamma, x \rangle > 0\}, \\
H_{\Gamma,i}^0 &:= \{x \in \mathbb{R}^d \colon \langle a_i^\Gamma, x \rangle = 0\}, \\
H_{\Gamma,i}^- &:= \{x \in \mathbb{R}^d \colon \langle a_i^\Gamma, x \rangle < 0\}.
\end{aligned} \tag{85}
$$

By conditions C.2 and C.3a we have for all $i, j \in I^\Gamma$ that

$$
i \neq j \implies H_{\Gamma,i}^0 \neq H_{\Gamma,j}^0. \tag{86}
$$

Further note that we can reparametrize $\mathcal{R}(\Theta)$ such that the same holds there. To this end observe that

$$
c\rho(\langle a, x \rangle) + c'\rho(\langle a', x \rangle) = (c + c' \tfrac{\|a'\|_\infty}{\|a\|_\infty})\rho(\langle a, x \rangle), \tag{87}
$$

given that $a'$ is a positive multiple of $a$. Specifically, let $(J_k)_{k=1}^K$ be a partition of $I^\Theta$ (i.e. $J_k \neq \emptyset$, $\cup_{k=1}^K J_k = I^\Theta$ and $J_k \cap J_{k'} = \emptyset$ if $k \neq k'$), such that for all $k \in [K]$ it holds that

$$
i, j \in J_k \implies \frac{a_j^\Theta}{\|a_j^\Theta\|_\infty} = \frac{a_i^\Theta}{\|a_i^\Theta\|_\infty}. \tag{88}
$$

We denote by $j_k$ the smallest element in $J_k$ and make the following replacements, for all $i \in I^\Theta$, without changing the realization of $\Theta$:

$$
a_i^\Theta \mapsto a_i^\Theta, c_i^\Theta \mapsto \sum_{j \in J_k} c_j^\Theta \frac{\|a_j^\Theta\|_\infty}{\|a_{j_k}^\Theta\|_\infty}, \quad \text{if } i \in J_k \text{ and } i = j_k, \tag{89}
$$

$$
a_i^\Theta \mapsto 0, c_i^\Theta \mapsto 0, \qquad\qquad\qquad \text{if } i \in J_k \text{ and } i \neq j_k. \tag{90}
$$

Note that we also update the set $I^\Theta := \{i \in [m] \colon a_i^\Theta \neq 0\}$ accordingly. Let now

$$
\begin{aligned}
H_{\Theta,i}^+ &:= \{x \in \mathbb{R}^d \colon \langle a_i^\Theta, x \rangle > 0\}, \\
H_{\Theta,i}^0 &:= \{x \in \mathbb{R}^d \colon \langle a_i^\Theta, x \rangle = 0\}, \\
H_{\Theta,i}^- &:= \{x \in \mathbb{R}^d \colon \langle a_i^\Theta, x \rangle > 0\}.
\end{aligned} \tag{91}
$$

By construction and condition C.3a, we have for all $i, j \in I^\Theta$ that

$$
i \neq j \implies H_{\Theta,i}^0 \neq H_{\Theta,j}^0. \tag{92}
$$

Note that we now have a parametrization $\Theta$ of $g$, where all weight vectors $a_i^\Theta$ are either zero (in which case the corresponding $c_i^\Theta$ are also zero) or pairwise linearly independent to each other nonzero weight vector.

Next, for $s \in \{0, 1\}^m$, let

$$
\begin{aligned}
H_\Gamma^s &:= \bigcap_{i \in [m] \colon s_i = 1} H_{\Gamma,i}^+ \cap \bigcap_{i \in [m] \colon s_i = 0} H_{\Gamma,i}^-, \\
H_\Theta^s &:= \bigcap_{i \in [m] \colon s_i = 1} H_{\Theta,i}^+ \cap \bigcap_{i \in [m] \colon s_i = 0} H_{\Theta,i}^-,
\end{aligned} \tag{93}
$$

and

$$
S^\Gamma := \{s \in \{0, 1\}^m \colon H_\Gamma^s \neq \emptyset\}, \quad S^\Theta := \{s \in \{0, 1\}^m \colon H_\Theta^s \neq \emptyset\}. \tag{94}
$$

The $H_\Gamma^s$, $s \in S^\Gamma$, and $H_\Theta^s$, $s \in S^\Theta$, are the interiors of the different linear regions of $\mathcal{R}(\Gamma)$ and $\mathcal{R}(\Theta)$ respectively. Next observe that the derivatives of $f_i^\Gamma, f_i^\Theta$ are (a.e.) given by

$$
Df_i^\Gamma(x) = \mathbf{1}_{H_{\Gamma,i}^+}(x)\, c_i^\Gamma a_i^\Gamma, \quad Df_i^\Theta(x) = \mathbf{1}_{H_{\Theta,i}^+}(x)\, c_i^\Theta a_i^\Theta. \tag{95}
$$

Note that for every $x \in H_\Gamma^s$, $y \in H_\Theta^s$ we have

$$
\begin{aligned}
D\mathcal{R}(\Gamma)(x) &= \sum_{i \in [m]} Df_i^\Gamma(x) = \sum_{i \in [m]} s_i c_i^\Gamma a_i^\Gamma =: \Sigma_s^\Gamma, \\
D\mathcal{R}(\Theta)(y) &= \sum_{i \in [m]} Df_i^\Theta(y) = \sum_{i \in [m]} s_i c_i^\Theta a_i^\Theta =: \Sigma_s^\Theta.
\end{aligned} \tag{96}
$$

Next we use that for $s \in S^\Gamma$, $t \in S^\Theta$ we have $|\Sigma_s^\Gamma - \Sigma_t^\Theta| \le r$ if $H_s^\Gamma \cap H_t^\Theta \ne \emptyset$, and compare adjacent linear regions of $\mathcal{R}(\Gamma) - \mathcal{R}(\Theta)$. Let now $i \in I^\Gamma$ and consider the following cases:

**Case 1**: We have $H_{\Gamma,i}^0 \ne H_{\Theta,j}^0$ for all $j \in I^\Theta$. This means that the $Df_k^\Theta$, $k \in [m]$, and the $Df_k^\Gamma$, $k \in [m]\backslash\{i\}$, are the same on both sides near the hyperplane $H_{\Gamma,i}^0$, while the value of $Df_i^\Gamma$ is 0 on one side and $c_i^\Gamma a_i^\Gamma$ on the other. Specifically, there exist $s^+, s^- \in S^\Gamma$ and $s^* \in S^\Theta$ such that $s_i^+ = 1$, $s_i^- = 0$, $s_j^+ = s_j^-$ for all $j \in [m]\backslash\{i\}$, and $H_\Gamma^{s^+} \cap H_\Theta^{s^*} \ne \emptyset$, $H_\Gamma^{s^-} \cap H_\Theta^{s^*} \ne \emptyset$, which implies

$$\|c_i^\Gamma a_i^\Gamma\|_\infty = \|(\Sigma_{s^+}^\Gamma - \Sigma_{s^*}^\Theta) - (\Sigma_{s^-}^\Gamma - \Sigma_{s^*}^\Theta)\|_\infty \le 2r. \tag{97}$$

**Case 2**: There exists $j \in I^\Theta$ such that $H_{\Gamma,i}^0 = H_{\Theta,j}^0$. Note that (86) ensures that $H_{\Gamma,i}^0 \ne H_{\Gamma,k}^0$ for $k \in [m] \backslash \{i\}$ and (92) ensures that $H_{\Theta,j}^0 \ne H_{\Theta,k}^0$ for $k \in [m] \backslash \{j\}$. Moreover, Condition C.3b implies $H_{\Gamma,i}^+ = H_{\Theta,j}^+$. This means that the $Df_k^\Theta$, $k \in [m]\backslash\{j\}$, and the $Df_k^\Gamma$, $k \in [m]\backslash\{i\}$, are the same on both sides near the hyperplane $H_{\Gamma,i}^0 = H_{\Theta,j}^0$, while the values of $Df_i^\Gamma$ and $Df_j^\Theta$ change. Specifically there exist $s^+, s^- \in S^\Gamma$ and $t^+, t^- \in S^\Theta$ such that $s_i^+ = 1$, $s_i^- = 0$, $s_k^+ = s_k^-$ for all $k \in [m]\backslash\{i\}$, $t_j^+ = 1$, $t_j^- = 0$, $t_k^+ = t_k^-$ for all $k \in [m]\backslash\{j\}$ and $H_{s^+}^\Gamma \cap H_{t^+}^\Theta \ne \emptyset$, $H_{s^-}^\Gamma \cap H_{t^-}^\Theta \ne \emptyset$, which implies

$$\|c_i^\Gamma a_i^\Gamma - c_j^\Theta a_j^\Theta\|_\infty = \|(\Sigma_{s^+}^\Gamma - \Sigma_{t^+}^\Theta) - (\Sigma_{s^-}^\Gamma - \Sigma_{t^-}^\Theta)\|_\infty \le 2r. \tag{98}$$

Analogously we get for $i \in I^\Theta$ that $H_{\Theta,i}^0 \ne H_{\Gamma,j}^0$ for all $j \in I^\Gamma$ implies $\|c_i^\Theta a_i^\Theta\|_\infty \le 2r$. Next let

$$I_1 := \{i \in [m]\colon H_{\Gamma,i}^0 \ne H_{\Theta,j}^0 \text{ for all } j \in I^\Theta\} \cup \{i \in [m]\colon a_i^\Gamma = 0\} \tag{99}$$

and

$$I_2 := [m] \backslash I_1 = \{i \in [m]\colon \exists\, j \in I^\Theta \text{ such that } H_{\Gamma,i}^+ = H_{\Theta,j}^+\}. \tag{100}$$

Colloquially speaking, this shows that for every $f_i^\Gamma$ with $i \in I_2$ there is a $f_j^\Theta$ with exactly matching half-spaces, i.e. $H_{\Gamma,i}^+ = H_{\Theta,j}^+$, and approximately matching gradients (Case 2). Moreover, all unmatched $f_i^\Gamma$ and $f_j^\Theta$ must have a small gradient (Case 1).

Specifically, the above establishes that there exists a permutation $\pi\colon [m] \to [m]$ such that for every $i \in I_1$ it holds that

$$\|c_i^\Gamma a_i^\Gamma\|_\infty, \|c_{\pi(i)}^\Theta a_{\pi(i)}^\Theta\|_\infty \le 2r, \tag{101}$$

and for every $i \in I_2$ that

$$\|c_i^\Gamma a_i^\Gamma - c_{\pi(i)}^\Theta a_{\pi(i)}^\Theta\|_\infty \le 2r. \tag{102}$$

We make the following replacements, for all $i \in [m]$, without changing the realization of $\Theta$:

$$a_i^\Theta \to a_{\pi(i)}^\Theta, \quad c_i^\Theta \to c_{\pi(i)}^\Theta. \tag{103}$$

In order to balance the weights of $\Theta$ for $I_1$, we further make the following replacements, for all $i \in I_1$ with $a_i^\Theta \ne 0$, without changing the realization of $\Theta$:

$$a_i^\Theta \to \left(\frac{|c_i^\Theta|}{\|a_i^\Theta\|_\infty}\right)^{1/2} a_i^\Theta, \quad c_i^\Theta \to \left(\frac{\|a_i^\Theta\|_\infty}{|c_i^\Theta|}\right)^{1/2} c_i^\Theta. \tag{104}$$

This implies for every $i \in I_1$ that

$$|c_i^\Theta|, \|a_i^\Theta\|_\infty \le (2r)^{1/2}. \tag{105}$$

Moreover, due to Condition C.1, we get for every $i \in I_1$ that

$$|c_i^\Gamma|, \|a_i^\Gamma\|_\infty \le \beta. \tag{106}$$

Thus we get for every $i \in I_1$ that

$$|c_i^\Theta - c_i^\Gamma|, \|a_i^\Theta - a_i^\Gamma\|_\infty \le \beta + (2r)^{1/2}. \tag{107}$$

Next we (approximately) match the balancing of $(c_i^\Theta, a_i^\Theta)$ to the balancing of $(c_i^\Gamma, a_i^\Gamma)$ for $i \in I_2$, in order to derive estimates on $|c_i^\Theta - c_i^\Gamma|$ and $\|a_i^\Theta - a_i^\Gamma\|_\infty$ from (102). Specifically, we make the following replacements, for all $i \in I_2$, without changing the realization of $\Theta$:

$$a_i^\Theta \to \left(\frac{|c_i^\Theta|}{\|a_i^\Theta\|_\infty}\right)^{1/2} a_i^\Theta, \quad c_i^\Theta \to \left(\frac{\|a_i^\Theta\|_\infty}{|c_i^\Theta|}\right)^{1/2} c_i^\Theta, \qquad \text{if } \|c_i^\Gamma a_i^\Gamma\|_\infty \le 2r, \tag{108}$$

$$a_i^\Theta \to \frac{c_i^\Theta}{c_i^\Gamma} a_i^\Theta, \quad c_i^\Theta \to c_i^\Gamma, \qquad \text{if } \|c_i^\Gamma a_i^\Gamma\|_\infty > 2r, |c_i^\Gamma| > \|a_i^\Gamma\|_\infty, \tag{109}$$

$$a_i^\Theta \to a_i^\Gamma, \quad c_i^\Theta \to \frac{\|a_i^\Theta\|_\infty}{\|a_i^\Gamma\|_\infty} c_i^\Theta, \qquad \text{if } \|c_i^\Gamma a_i^\Gamma\|_\infty > 2r, |c_i^\Gamma| < \|a_i^\Gamma\|_\infty, \tag{110}$$

$$a_i^\Theta \to \left(\frac{|c_i^\Theta|}{\|a_i^\Theta\|_\infty}\right)^{1/2} a_i^\Theta, \quad c_i^\Theta \to \left(\frac{\|a_i^\Theta\|_\infty}{|c_i^\Theta|}\right)^{1/2} c_i^\Theta, \quad \text{if } \|c_i^\Gamma a_i^\Gamma\|_\infty > 2r, |c_i^\Gamma| = \|a_i^\Gamma\|_\infty. \tag{111}$$

Let now $i \in I_2$ and consider the following cases:

**Case A**: We have $\|c_i^\Gamma a_i^\Gamma\|_\infty \le 2r$ which, together with (102), implies $\|c_i^\Theta a_i^\Theta\|_\infty \le 4r$. Due to (108) and Condition C.1 it follows that

$$|c_i^\Theta - c_i^\Gamma|, \|a_i^\Theta - a_i^\Gamma\|_\infty \le \beta + 2r^{1/2}. \tag{112}$$

**Case B.1**: We have $\|c_i^\Gamma a_i^\Gamma\|_\infty > 2r$ and $|c_i^\Gamma| > \|a_i^\Gamma\|_\infty$ which ensures $|c_i^\Gamma| > \|c_i^\Gamma a_i^\Gamma\|_\infty^{1/2}$. Due to (109) we get $c_i^\Theta = c_i^\Gamma$ and it follows that

$$\|a_i^\Theta - a_i^\Gamma\|_\infty = \frac{1}{|c_i^\Gamma|} \|c_i^\Theta a_i^\Theta - c_i^\Gamma a_i^\Gamma\|_\infty \le \frac{2r}{\|c_i^\Gamma a_i^\Gamma\|_\infty^{1/2}} \le (2r)^{1/2}. \tag{113}$$

**Case B.2**: We have $\|c_i^\Gamma a_i^\Gamma\|_\infty > 2r$ and $|c_i^\Gamma| < \|a_i^\Gamma\|_\infty$ which ensures $\|a_i^\Gamma\| > \|c_i^\Gamma a_i^\Gamma\|_\infty^{1/2}$. Due to (110) we get $a_i^\Theta = a_i^\Gamma$ and it follows that

$$|c_i^\Theta - c_i^\Gamma| = \frac{1}{\|a_i^\Gamma\|_\infty} \|c_i^\Theta a_i^\Theta - c_i^\Gamma a_i^\Gamma\|_\infty \le \frac{2r}{\|c_i^\Gamma a_i^\Gamma\|_\infty^{1/2}} \le (2r)^{1/2}. \tag{114}$$

**Case B.3**: We have $\|c_i^\Gamma a_i^\Gamma\|_\infty > 2r$ and $|c_i^\Gamma| = \|a_i^\Gamma\|_\infty$. Note that $\|c_i^\Gamma a_i^\Gamma\|_\infty > 2r$ and (102) ensure that $\mathrm{sgn}(c_i^\Theta) = \mathrm{sgn}(c_i^\Gamma)$, and that for $x, y > 0$ it holds that $|x - y| \le |x^2 - y^2|^{1/2}$. Combining this with the definition of $I_2$, the reverse triangle inequality, and (111) implies that

$$\|a_i^\Theta - a_i^\Gamma\|_\infty \le (2r)^{1/2} \quad \text{and} \quad |c_i^\Theta - c_i^\Gamma| \le (2r)^{1/2}. \tag{115}$$

Combining (107), (112), (113), (114), and (115) establishes that

$$\|\Theta - \Gamma\|_\infty \le \beta + 2r^{\frac{1}{2}}, \tag{116}$$

which completes the proof. $\qquad \square$

*Proof of Theorem 3.3.* Let $\Theta \in \mathcal{N}_N^*$ be a parametrization of $g$, i.e. $\mathcal{R}(\Theta) = g$. We write

$$\Gamma = \left( \begin{bmatrix} a_1^\Gamma \\ \vdots \\ a_m^\Gamma \end{bmatrix}, [c_1^\Gamma | \dots | c_m^\Gamma] \right), \quad \Theta = \left( \begin{bmatrix} a_1^\Theta \\ \vdots \\ a_m^\Theta \end{bmatrix}, [c_1^\Theta | \dots | c_m^\Theta] \right) \in \mathcal{N}_{(d,m,D)}^* \tag{117}$$

and $r := |g - \mathcal{R}(\Gamma)|_{W^{1,\infty}}$. For convenience of notation we consider the weight vectors $a_i^\Gamma$, $a_i^\Theta$ here as row vectors in order to write the derivatives of the ridge functions as $c_i^\Gamma a_i^\Gamma$, $c_i^\Theta a_i^\Theta \in \mathbb{R}^{D \times d}$ without transposing.

We will now adjust the approach used in the proof of Theorem 3.1 to work for multi-dimensional outputs in the case of balanced networks. By definition of $\mathcal{N}_N^*$, the $(a_i^\Theta)_{i=1}^m$ are pairwise linearly independent and we can skip the first reparametrization step in (89) and (90).

The following "hyperplane-jumping" argument, which was used to get the estimates (97) and (98), works analogously since Conditions C.2 and C.3 are fulfilled by definition of $\mathcal{N}_N^*$. This establishes the existence of a permutation $\pi \colon [m] \to [m]$ and sets $I_1, I_2 \subseteq [m]$, as defined as in (99) and (100), such that for every $i \in I_1$ it holds that

$$\|c_i^\Gamma a_i^\Gamma\|_\infty, \|c_{\pi(i)}^\Theta a_{\pi(i)}^\Theta\|_\infty \le 2r, \tag{118}$$

and for every $i \in I_2$ that

$$\|c_i^\Gamma a_i^\Gamma - c_{\pi(i)}^\Theta a_{\pi(i)}^\Theta\|_\infty \leq 2r. \tag{119}$$

As in (103), we make the following replacements, for all $i \in [m]$, without changing the realization of $\Theta$:

$$a_i^\Theta \to a_{\pi(i)}^\Theta, \quad c_i^\Theta \to c_{\pi(i)}^\Theta. \tag{120}$$

Note that the weights of $\Theta$ are already balanced, i.e. we have for every $i \in [m]$ that

$$\|c_i^\Theta\|_\infty = \|a_i^\Theta\|_\infty = \|c_i^\Theta\|_\infty^{1/2}\|a_i^\Theta\|_\infty^{1/2} = \|c_i^\Theta a_i^\Theta\|_\infty^{1/2}. \tag{121}$$

Thus, we can skip the reparametrization step in (104) and get directly for every $i \in I_1$ that

$$\|c_i^\Theta - c_i^\Gamma\|_\infty \leq \|c_i^\Theta\|_\infty + \|c_i^\Gamma\|_\infty = \|c_i^\Theta a_i^\Theta\|_\infty^{1/2} + \|c_i^\Gamma a_i^\Gamma\|_\infty^{1/2} \leq 2(2r)^{1/2} \tag{122}$$

and analogously $\|a_i^\Theta - a_i^\Gamma\|_\infty \leq 2(2r)^{1/2}$.

For $i \in I_2$ we need to slightly deviate from the proof of Theorem 3.1. We can skip the reparametrization step in (108)-(111) due to balancedness and need to distinguish three cases:

**Case A.1**: We have $\|c_i^\Gamma a_i^\Gamma\|_\infty \leq 2r$ which, together with (119), implies $\|c_i^\Theta a_i^\Theta\|_\infty \leq 4r$. Due to balancedness it follows that

$$\|c_i^\Theta - c_i^\Gamma\|_\infty, \|a_i^\Theta - a_i^\Gamma\|_\infty \leq 4r^{1/2}. \tag{123}$$

**Case A.2**: We have $\|c_i^\Theta a_i^\Theta\|_\infty \leq 2r$ which, together with (119), implies $\|c_i^\Gamma a_i^\Gamma\|_\infty \leq 4r$. Again it follows that

$$\|c_i^\Theta - c_i^\Gamma\|_\infty, \|a_i^\Theta - a_i^\Gamma\|_\infty \leq 4r^{1/2}. \tag{124}$$

**Case B**: We have $\|c_i^\Theta a_i^\Theta\|_\infty > 2r$ and $\|c_i^\Gamma a_i^\Gamma\|_\infty > 2r$. Due to the definition of $I_2$ there exists $e_i \in \mathbb{R}^d$, $\lambda_i^\Gamma, \lambda_i^\Theta \in (0, \infty)$ with $\|e_i\|_\infty = 1$, $a_i^\Theta = \lambda_i^\Theta e_i$, and $a_i^\Gamma = \lambda_i^\Gamma e_i$. As in (115) we obtain that

$$\begin{aligned}
\|a_i^\Theta - a_i^\Gamma\|_\infty &= \|e_i\|_\infty |\lambda_i^\Theta - \lambda_i^\Gamma| \leq |(\lambda_i^\Theta)^2 - (\lambda_i^\Gamma)^2|^{1/2} \\
&= |\|c_i^\Theta\|_\infty \|a_i^\Theta\|_\infty - \|c_i^\Gamma\|_\infty \|a_i^\Gamma\|_\infty|^{1/2} \\
&\leq \|c_i^\Theta a_i^\Theta - c_i^\Gamma a_i^\Gamma\|_\infty^{1/2} \leq (2r)^{1/2}.
\end{aligned} \tag{125}$$

Let now w.l.o.g. $\|a_i^\Gamma\|_\infty \geq \|a_i^\Theta\|_\infty$ (otherwise we switch their roles in the following) which implies that $\lambda_i^\Gamma = \Delta_i + \lambda_i^\Theta$ with $\Delta_i = \lambda_i^\Gamma - \lambda_i^\Theta \geq 0$. Then it holds that

$$\begin{aligned}
\|c_i^\Theta - c_i^\Gamma\|_\infty &= \frac{\|c_i^\Theta a_i^\Gamma - c_i^\Gamma a_i^\Gamma\|_\infty}{\|a_i^\Gamma\|_\infty} \leq \frac{\|c_i^\Theta a_i^\Gamma - c_i^\Theta a_i^\Theta\|_\infty + \|c_i^\Theta a_i^\Theta - c_i^\Gamma a_i^\Gamma\|_\infty}{\|a_i^\Gamma\|_\infty} \\
&\leq \frac{\|c_i^\Theta\|_\infty |\lambda_i^\Gamma - \lambda_i^\Theta| + 2r}{\lambda_i^\Gamma} = \frac{\lambda_i^\Theta \Delta_i + 2r}{\Delta_i + \lambda_i^\Theta} \\
&= \frac{(2r)^{1/2}(\Delta_i + \lambda_i^\Theta) - (\lambda_i^\Theta - (2r)^{1/2})((2r)^{1/2} - \Delta_i)}{\Delta_i + \lambda_i^\Theta} \leq (2r)^{1/2}.
\end{aligned} \tag{126}$$

The last step holds due to (125) and the balancedness of $\Theta$ which ensure that

$$\lambda_i^\Theta = \|c_i^\Theta a_i^\Theta\|_\infty^{1/2} > (2r)^{1/2} \geq |\lambda_i^\Theta - \lambda_i^\Gamma| = \Delta_i. \tag{127}$$

This completes the proof. $\square$

## A.4 Section 4

### A.4.1 Additional Material

**Lemma A.7** (Inverse stability for fixed weight vectors). *Let $N = (d, m, D) \in \mathbb{N}^3$, let $A = [a_1| \ldots |a_m]^T \in \mathbb{R}^{m \times d}$ with*

$$\frac{a_i}{\|a_i\|_\infty} \neq \frac{a_j}{\|a_j\|_\infty} \quad and \quad (a_i)_{d-1}, (a_i)_d > 0 \tag{128}$$

*for all $i \in [m], j \in [m] \setminus \{i\}$, and define*

$$\mathcal{N}_N^A := \left\{ \Gamma \in \mathcal{N}_N \colon a_i^\Gamma = \lambda_i a_i \text{ with } \lambda_i \in (0, \infty) \text{ and } \|c_i^\Gamma\|_\infty = \|a_i^\Gamma\|_\infty \text{ for all } i \in [m] \right\}. \quad (129)$$

*Then for every $B \in (0, \infty)$ there is $C_B \in (0, \infty)$ such that we have uniform $(C_B, 1/2)$ inverse stability w.r.t. $\|\cdot\|_{L^\infty((-B,B)^d)}$. That is, for all $\Gamma \in \mathcal{N}_N^A$ and $g \in \mathcal{R}(\mathcal{N}_N^A)$ there exists a parametrization $\Phi \in \mathcal{N}_N^A$ with*

$$\mathcal{R}(\Phi) = g \quad \text{and} \quad \|\Phi - \Gamma\|_\infty \le C_B \|g - \mathcal{R}(\Gamma)\|_{L^\infty((-B,B)^d)}^{\frac{1}{2}}. \quad (130)$$

*Proof.* Note that the non-zero angle between the hyperplanes given by the weight vectors $(a_i)_{i=1}^m$ establishes that the minimal perimeter inside each linear region intersected with $(-B, B)^d$ is lower bounded. As the realization is linear on each region, this implies the existence of a constant $C_B' \in (0, \infty)$, such that for every $\Theta \in \mathcal{N}_N^A$ it holds that

$$|\mathcal{R}(\Theta)|_{W^{1,\infty}} \le C_B' \|\mathcal{R}(\Theta)\|_{L^\infty((-B,B)^d)}. \quad (131)$$

Now note that for $\mathcal{N}_N^A$ we can get the same uniform $(4, 1/2)$ inverse stability result w.r.t. $|\cdot|_{W^{1,\infty}}$ as in Theorem 3.3 by choosing $\pi$ to be the identity in (118). Together with (131) this implies the claim. $\qquad\square$

## Footnotes

[5]See notation in the beginning of Section 2.

[6] In case one of them is zero, the other one can be set to zero without changing the realization.