[Reviews · NeurIPS 2019]

Reviewer 1



The paper follows on a previous distinction between the weights of a neural network and the function that is realizes, and investigates under what conditions can we guarantee inverse stability: That networks with a similar realization will also be close in the parameter space (or more accurately, whether there exists an equivalent realization of one network that is close in the parameter space to the other). First, the authors identify that inverse stability fails w.r.t. the uniform norm, which motivates the investigation of inverse stability w.r.t. the Sobolev norm. Thereafter, the authors identify 4 cases where inverse stability fails even w.r.t the Sobolev norm, and lastly they show that under suitable assumptions that eliminate these 4 pathologies, inverse stability w.r.t. to the Sobolev norm for one hidden layer networks is achieved, where they argue (without proof) their technique also extends to deeper architectures. The authors motivate this line of research by establishing a correspondence between minima in the optimization space and minima in the realization space, and arguing that this connection will allow the adaptation of known tools from e.g. functional analysis to investigate neural networks. While I find the paper well-written and clear, my main concern is with its novelty, as specified in the improvements part. Minor comments: Line 22: year -> years. Line 72: "The, to the best..." is unclear. Possible typo? Line 218: an restriction -> a restriction

Reviewer 2



I read the author response and other reviews. The author response provides nice additional demonstration about the implication of connecting the two problems via inverse stability. This is an interesting and potentially important paper for a future research on this topic. This paper explains the definition of the inverse stability, proves its implication for neural network optimization, provides failure modes of having the inverse stability, and proves the inverse stability for a simple one-hidden layer network with a single output. Originality: The paper definitely provides a very interesting and unique research direction. I enjoyed reading the paper. It is quite refreshing to have a paper that acknowledges the limitation of the current theories (very strong assumptions) and aims to go beyond the regime of the strong assumptions. The present paper is related to a previous paper that concludes a global minimum of some space from local minimum of the parameter space. Although it is different, the goal is closely related and the high-level approach to relate the original problem to another is the same. In the present paper, the parameter local minimum only implies the realization local minimum, whereas the related paper concludes the global minimum. However, the realization space is richer and the present paper is good and interesting. Quality: The quality of the paper is good. Clarity: paper is clearly written. Significance: I found the paper interesting and enjoy reading it. A concern that I could think of is that the results are currently limited to the very simple case. I think, this is understandable due to the challenging nature of the topic. One may consider that this can be a concern because of the following: even if we have a complete picture about the inverse stability, that would be only the beginning of studying the neural network in the realization form. The statement about the possibility of the extension to deep neural networks is unnecessary and tautological. Of course, it is possible, but it is highly non-trivial and at what cost? For example, it may end up requiring the strong assumptions that this paper wanted to avoid.

Reviewer 3



The conditions of main Theorem 3.1 are quite technical and hard to interpret. It is not clear to me what this result actually implies in practice (also, on the original problem) and how it can be helpful in analyzing or understanding better the training of neural networks. Although this is motivated earlier in Corollary 1.3, again, the conditions there are quite technical and the results does not seem to say much about the properties of local minima of the original problem.

[Author Response · NeurIPS 2019]

We thank the reviewers for the feedback and will address their concerns in the following.

**Motivation and significance of our research direction:** (reviewer 1,2,3)

Below we present a result obtained on the realization space, which shows that for sufficiently large architectures all
local minima of a regularized neural network optimization problem are almost optimal. We then use inverse stability to
translate this result to the practically relevant parametrized problem. As suggested by reviewer 1 we will include this in
the camera-ready version.

For the following we fix a depth $L$ as well as input/output dimensions $N_0, N_L$. We denote by $\mathcal{A}$ the set of all architectures
$N = (N_0, N_1, \ldots, N_L)$ with this depth and these input/output dimensions, and by $\mathcal{P}$ the set of all parametrizations
with architecture in $\mathcal{A}$. Let $(X, \|\cdot\|)$ be a Banach space with $\mathcal{R}(\mathcal{P}) \subseteq X$ and let $\Lambda\colon X \mapsto \mathbb{R}_+$ be a quasi-convex
regularizer. Define $S := \{f \in X\colon \Lambda(f) \leq C\}$ and assume that S is compact in the $\|\cdot\|$-closure of $\mathcal{R}(\mathcal{P})$. We denote
the sets of regularized parametrizations by $\Omega_N := \{\Phi \in \mathcal{P}_N\colon \Lambda(\mathcal{R}(\Phi)) \leq C\}$ and consider a convex and $c$-Lipschitz
loss function $\mathcal{L}$ on $S$ (note that this is fulfilled for virtually all relevant loss functions).

**Theorem 1** (Almost optimality of local realization minima). *For all $\varepsilon, r > 0$ there exists $n(\varepsilon, r) \in \mathcal{A}$ such that for*
*every $N \in \mathcal{A}$ with $N_1 \geq n_1(\varepsilon, r), \ldots, N_{L-1} \geq n_{L-1}(\varepsilon, r)$ the following holds:*
*Every local minimum $h_*$ with radius at least $r$ of the optimization problem $\min_{h \in \mathcal{R}(\Omega_N)} \mathcal{L}(h)$ satisfies*

$$\mathcal{L}(h_*) \leq \min_{h \in \mathcal{R}(\Omega_N)} \mathcal{L}(h) + \varepsilon.$$

*Proof.* Let $\eta := \min\left\{\frac{r\varepsilon}{2c\,\mathrm{diam}(S)}, \frac{r}{2}\right\}$. Due to compactness of $S$ there exists $n(\varepsilon, r) \in \mathcal{A}$ such that for every $N \in \mathcal{A}$
with $N_1 \geq n_1(\varepsilon, r), \ldots, N_{L-1} \geq n_{L-1}(\varepsilon, r)$ it holds that $\sup_{f \in S} \inf_{\Phi \in \Omega_N} \|\mathcal{R}(\Phi) - f\| \leq \eta$. Let $h \in \mathcal{R}(\Omega_N)$ and
define $\lambda := \frac{r}{2\|h - h_*\|}$ and $f := (1 - \lambda)h_* + \lambda h \in S$. By the assumptions on $h_*$ and $\mathcal{L}$ it holds that

$$\mathcal{L}(h_*) \leq \mathcal{L}(\mathcal{R}(\Phi)) \leq \mathcal{L}(f) + c\eta \leq (1 - \lambda)\mathcal{L}(h_*) + \lambda\mathcal{L}(h) + c\eta.$$

Direct computation establishes the claim. $\qquad\qquad\qquad\qquad\qquad\qquad\qquad\qquad\qquad\qquad\qquad\qquad$ □

Now inverse stability is necessary (see Example A.1 in the paper) and sufficient (see Prop. 1.2 in the paper) in order to
get the following corollary for the parametrized problem.

**Corollary 2** (Almost optimality of local parameter minima). *Assume that the realization map is $(s, \alpha)$ inverse stable*
*on $\Omega_N$ w.r.t $\|\cdot\|$ for every $N \in \mathcal{A}$. Then for all $\varepsilon, r > 0$ there exists $n(\varepsilon, r) \in \mathcal{A}$ such that for every $N \in \mathcal{A}$ with*
*$N_1 \geq n_1(\varepsilon, r), \ldots, N_{L-1} \geq n_{L-1}(\varepsilon, r)$ the following holds:*
*Every local minimum $\Gamma_*$ with radius at least $r$ of the optimization problem $\min_{\Gamma \in \Omega_N} \mathcal{L}(\mathcal{R}(\Gamma))$ satisfies*

$$\mathcal{L}(\mathcal{R}(\Gamma_*)) \leq \min_{\Gamma \in \Omega_N} \mathcal{L}(\mathcal{R}(\Gamma)) + \varepsilon.$$

*Proof.* $\mathcal{R}(\Gamma_*)$ is a local minimum with radius $(\frac{r}{s})^{1/\alpha}$ (Prop. 1.2 in the paper) and Theorem 1 implies the claim.   □

Note that it is important to have an inverse stability result, where the $(s, \alpha)$ does not depend on the size of the
architecture, which we achieve in our submission for $L = 2$ and $X = W^{1,\infty}$. Suitable $\Lambda$ would be Besov norms which
constitute a common regularizer in image and signal processing. Moreover note that the required size of the architecture
in Theorem 1 and Corollary 2 can be quantified, if one has approximation rates for $S$. In particular, this approach
allows the use of approximation results in order to explain the success of neural network optimization and allows for a
combined study of these two aspects, which, to the best of our knowledge, has not been done before. Unlike in recent
literature, our result needs no assumptions on the sample set (incorporated in the loss function, as shown in the paper),
in particular we do not require 'overparametrization' with respect to the sample size. Here the required size of the
architecture only depends on the complexity of $S$, i.e. the class of functions one wants to approximate, the radius of the
local minima of interest, the Lipschitz constant of the loss function, and the parameters of the inverse stability.

**Extension to deep architectures and multiple output units:** (reviewer 2)

We have the extension to multiple output units worked out (it requires adjusting the notion of balancedness but otherwise
follows directly using techniques from the paper) and will include it in the final version. While a full extension to deep
networks would exceed the scope of the submission, we will add a discussion of the challenges and possible solutions.

**Motivation for the non-degeneracy conditions of Theorem 3.1 and contribution to regularization:** (reviewer 3)

We would like to highlight the following additional merit of the study of degenerate parametrizations. Without inverse
stability it is possible to get stuck in a local parameter minimum that is not a realization minimum, which could be
prevented by regularizations designed to exclude the pathologies we found. In order to make the technical conditions in
Theorem 3.1 in the paper more palatable we will link them to practical methods of regularization. Specifically, we will
cite and comment on recent NeurIPS resp. ICLR papers where the authors achieved promising empirical results by
using a regularization term (e.g. based on cosine similarity) corresponding to conditions C.2 and C.3 of the theorem.

We think this addresses all the points of criticism, regarding motivation, results on neural network optimization, and
extensions to (deep) architectures with multiple output units and we will include these improvements in the final version.

[Meta-Review · NeurIPS 2019]

The paper provides interesting new theoretical results on inverse stability. As with all current theory papers about neural networks, it is unclear whether this research direction will prove fruitful for practical insights, and this uncertainty is reflected in the reviews. However, the paper has sufficient redeeming qualities to merit acceptance: the results are clean, the paper is very well written, and the rebuttal provides sufficiently substantial indications of how the results can have an impact. Further remarks: * I agree with reviewer #2 that the claim about extensions to deeper networks seems unnecessary and detracts from the paper. You might replace this by a discussion of the potential difficulties of such an extension. * For future work, I would suggest to the authors that establishing the importance of the research direction might have higher priority than generalizing the results to deeper networks.